# Do LLMs dream of elephants (when told not to)? Latent concept association and associative memory in transformers

Yibo Jiang[1], Goutham Rajendran[2], Pradeep Ravikumar[2], and Bryon Aragam[3]

[1]*Department of Computer Science, University of Chicago*
[2]*Machine Learning Department, Carnegie Mellon University*
[3]*Booth School of Business, University of Chicago*

## Abstract

Large Language Models (LLMs) have the capacity to store and recall facts. Through experimentation with open-source models, we observe that this ability to retrieve facts can be easily manipulated by changing contexts, even without altering their factual meanings. These findings highlight that LLMs might behave like an associative memory model where certain tokens in the contexts serve as clues to retrieving facts. We mathematically explore this property by studying how transformers, the building blocks of LLMs, can complete such memory tasks. We study a simple latent concept association problem with a one-layer transformer and we show theoretically and empirically that the transformer gathers information using self-attention and uses the value matrix for associative memory.

## 1 Introduction

What is the first thing that would come to mind if you were asked *not* to think of an elephant? Chances are, you would be thinking about elephants. What if we ask the same thing to Large Language Models (LLMs)? Obviously, one would expect the outputs of LLMs to be heavily influenced by tokens in the context [Bro+20]. Could such influence potentially prime LLMs into changing outputs in a nontrivial way? To gain a deeper understanding, we focus on one specific task called fact retrieval [Men+22; Men+23] where expected output answers are given. LLMs, which are trained on vast amounts of data, are known to have the capability to store and recall facts [Men+22; Men+23; DCAT21; Mit+21; Mit+22; Dai+21]. This ability raises natural questions: *How robust is fact retrieval, and to what extent does it depend on semantic meanings within contexts? What does it reveal about memory in LLMs?*

In this paper, we first demonstrate that fact retrieval is not robust and LLMs can be easily fooled by varying contexts. For example, when asked to complete "The Eiffel Tower is in the city of", GPT-2 [Rad+19] answers with "Paris". However, when prompted with "The Eiffel Tower is not in Chicago. The Eiffel Tower is in the city of", GPT-2 responds with "Chicago". See Figure 1 for more examples, including Gemma and LLaMA. On the other hand, humans do not find the two sentences factually confusing and would answer "Paris" in both cases. We call this phenomenon *context hijacking*. Importantly, these findings suggest that LLMs might behave like an associative memory model. Specifically, we refer to an associative memory model in which LLMs rely on certain tokens in contexts to guide the retrieval of memories, even if such associations formed are not inherently semantically meaningful. This contrasts with the ideal behavior, where LLMs would generalize by understanding new contexts, reasoning through them, and integrating prior knowledge.

This associative memory perspective raises further interpretability questions about how LLMs form such associations. Answering these questions can facilitate the development of more robust LLMs.

38th Conference on Neural Information Processing Systems (NeurIPS 2024).

| MODEL | CONTEXT | NEXT TOKEN |
|---|---|---|
| All models | The Eiffel Tower is in the city of | Paris |
| GPT-2 / Gemma-2B | The Eiffel Tower is not in Chicago. Therefore, the Eiffel Tower is in the city of | Chicago |
| Gemma-2B-IT | The Eiffel Tower is not in Chicago. However, the Chicago river is in Chicago. Therefore, the Eiffel Tower is in the city of | Chicago |
| LLaMA-7B | The Eiffel Tower is not in Chicago. The Eiffel Tower is not in Chicago. The Eiffel Tower is not in Chicago. The Eiffel Tower is not in Chicago. The Eiffel Tower is not in Chicago. The Eiffel Tower is not in Chicago. The Eiffel Tower is not in Chicago. Therefore, the Eiffel Tower is in the city of | Chicago |

**Figure 1:** Examples of context hijacking for various LLMs, showcasing that fact retrieval is not robust.

Unlike classical models of associative memory in which distance between memory patterns are measured directly and the associations between inputs and outputs are well-specified, fact retrieval relies on a more nuanced notion of similarity measured by latent (unobserved) semantic concepts. To model this, we propose a synthetic task called *latent concept association* where the output token is closely related to sampled tokens in the context but wherein similarity is measured via a latent space of semantic concepts. We then investigate how a one-layer transformer [Vas+17], a fundamental component of LLMs, can tackle this memory retrieval task in which various context distributions correspond to distinct memory patterns. We demonstrate that the transformer accomplishes the task in two stages: The self-attention layer gathers information, while the value matrix functions as associative memory. Moreover, low-rank structure also emerges in the embedding space of trained transformers. These findings provide additional theoretical validation for numerous existing low-rank editing and fine-tuning techniques [Men+22; Hu+21].

**Contributions**  Specifically, we make the following contributions:

1. We systematically demonstrate context hijacking for various open source LLM models including GPT-2 [Rad+19], LLaMA-2 [Tou+23] and Gemma [Tea+24], which show that fact retrieval can be misled by contexts (Section 3), reaffirming that LLMs lack robustness to context changes [Shi+23; Pet+20; CSH22; Yor+23; PE21].

2. We propose a synthetic memory retrieval task termed latent concept association, allowing us to analyze how transformers can accomplish memory recall (Section 4). Unlike classical models of associative memory, our task creates associations in a latent, semantic concept space as opposed to directly between observed tokens. This perspective is crucial to understanding how transformers can solve fact retrieval problems by implementing associative memory based on similarity in the latent space.

3. We theoretically (Section 5) and empirically (Section 6) study trained transformers on this latent concept association problem, showing that self-attention is used to aggregate information while the value matrix serves as associative memory. And moreover, we discover that the embedding space can exhibit a low-rank structure, offering additional support for existing editing and fine-tuning methods [Men+22; Hu+21].

## 2   Literature review

**Associative memory**  Associative memory has been explored within the field of neuroscience [Hop82; Seu96; BYBOS95; Ska+94; SS22]. The most popular models among them is the Hopfield network [Hop82] and its modern successors [Ram+20; Mil+22; Zha23; Hu+24d; Wu+23; Hu+24b; Hu+24c; Wu+24a; Hu+24a] are closely related to the attention layer used in transformers [Vas+17]. In addition, the attention mechanism has also been shown to approximate another associative memory model known as sparse distributed memory [BP21]. Beyond attention, Radhakrishnan et al. [RBU20] and Jiang and Pehlevan [JP20] show that overparameterized autoencoders can implement associative memory as well. This paper studies fact retrieval as a form of associative memory. Another closely

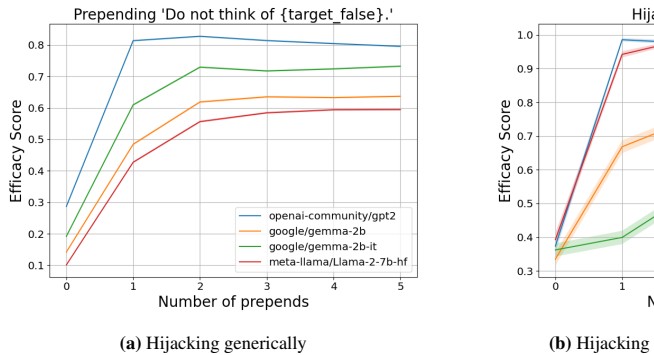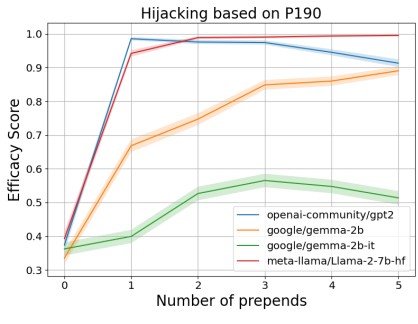

(a) Hijacking generically

(b) Hijacking based on Relation ID P190

**Figure 2:** Context hijacking can cause LLMs to output false target. The figure shows efficacy score versus the number of prepends for various LLMs on the COUNTERFACT dataset under two hijacking schemes.

related area of research focuses on memorization in deep neural networks. Henighan et al. [Hen+23] shows that a simple neural network trained on toy model will store data points in the overfitting regime while storing features in the underfitting regime. Feldman [Fel20] and Feldman and Zhang [FZ20] study the interplay between memorization and long tail distributions while Kim et al. [KKM22] and Mahdavi et al. [MLT23] study the memorization capacity of transformers.

**Interpreting transformers and LLMs** There's a growing body of work on understanding how transformers and LLMs work [LLR23; AZL23a; AZL23b; AZL24; EI+24; Tar+23b; Tar+23a; Li+24], including training dynamics [Tia+23a; Tia+23b; She+24] and in-context learning [Xie+21; Gar+22; Bai+24; Bai+24]. Recent papers have introduced synthetic tasks to better understand the mechanisms of transformers [Cha22; Liu+22; Nan+23; Zha+22; Zho+24], such as those focused on Markov chains [Bie+24; Ede+24; NDL24; Mak+24]. Most notably, Bietti et al. [Bie+24] and subsequent works [CDB23; CSB24] study weights in transformers as associative memory but their focus is on understanding induction head [Ols+22b] and one-to-one map between input query and output memory. An increasing amount of research is dedicated to understanding the internals of pre-trained LLMs, broadly categorized under the term "mechanistic interpretability" [Elh+21; Ols+22a; Gev+23; Men+22; Men+23; Jia+24; Raj+24; Has+24; Wan+22; McG+23; Gei+21; Gei+22; Gei+24; Wu+24b].

**Knowledge editing and adversarial attacks on LLMs** Fact recall and knowledge editing have been extensively studied [Men+22; Men+23; Has+24; Sak+23; DCAT21; Mit+21; Mit+22; Dai+21; Zha+23; Tia+24; Jin+23], including the use of in-context learning to edit facts [Zhe+23]. This paper aims to explore a different aspect by examining the robustness of fact recall to variation in prompts. A closely related line of work focuses on adversarial attacks on LLMs [see Cho+24, for a review]. Specifically, prompt-based adversarial attacks [Xu+23; Zhu+23; Wan+23b] focus on the manipulation of answers within specific classification tasks while other works concentrate on safety issues [Liu+23a; PR22; Zou+23; Apr+22; Wan+23a; Si+22; Rao+23; SMR23; Liu+23b]. Yu et al. [Yu+24] and Luo et al. [Luo+24] also study jailbreak phenomena within the context of modern Hopfield network. There are also works showing LLMs can be distracted by irrelevant contexts in problem solving [Shi+23], question answering [Pet+20; CSH22; Yor+23] and factual reasoning [PE21]. Although phenomena akin to context hijacking have been reported in different instances, the goals of this paper are to give a systematic robustness study for fact retrieval, offer a framework for interpreting it in the context of associative memory, and deepen our understanding of LLMs.

## 3 Context hijacking in LLMs

In this section, we run experiments on LLMs including GPT-2 [Rad+19], Gemma [Tea+24] (both base and instruct models) and LLaMA-2-7B [Tou+23] to explore the effects of context hijacking on manipulating LLM outputs. As an example, consider Figure 1. When we prompt the LLMs with the context "The Eiffel Tower is in the city of", all 4 LLMs output the correct answer ("Paris"). However, as we see in the example, we can actually manipulate the output of the LLMs simply by modifying the context with additional *factual* information that would not confuse a human. We call

this *context-hijacking*. Due to the different capacities and capabilties of each model, the examples in Figure 1 use different hijacking techniques. This is most notable on LLaMA-2-7B, which is a much larger model than the others. Of course, as expected, the more sophisticated attack on LLaMA also works on GPT-2 and Gemma. Additionally, the instruction-tuned version of Gemma can understand special words like "not" to some extent. Nevertheless, it is still possible to systematically hijack these LLMs, as demonstrated below.

We explore this phenomenon at scale with the COUNTERFACT dataset introduced in [Men+22], a dataset of difficult counterfactual assertions containing a diverse set of subjects, relations, and linguistic variations. COUNTERFACT has $21,919$ samples, each of which are given by a tuple $(p, o_*, o_-, s, r)$. From each sample, we have a context prompt $p$ with a true target answer $o_*$ (target_true) and a false target answer $o_-$ (target_false), e.g. the prompt $p =$ "Eiffel Tower can be found in" has true target $o_* =$ "Paris" and false target $o_- =$ "Guam". Additionally, the main entity in $p$ is the subject $s$ ($s =$ "Eiffel Tower") and the prompt is categorized into relations $r$ (for instance, other samples with the same relation ID as the example above could be of the form "The location of {subject} is", "{subject} can be found in", "Where is {subject}? It is in"). For additional details on how the dataset was collected, see [Men+22].

For a hijacking scheme, we report the Efficacy Score (ES) [Men+22], which is the proportion of samples for which the token probabilities satisfy $Pr[o_-] > Pr[o_*]$ after modifying the context, that is, the proportion of the dataset that has been successfully manipulated. We experiment with two hijacking schemes for this dataset. We first hijack by prepending the text "Do not think of {target_false}" to each context. For instance, the prompt "The Eiffel Tower is in" gets changed to "Do not think of Guam. The Eiffel Tower is in". In Figure 2a, we see that the efficacy score rises significantly after hijacking. Here, we prepend the hijacking sentence $k$ times for $k = 0, \ldots, 5$ where $k = 0$ yields the original prompt. We see that additional prepends increase the score further.

In the second scheme, we make use of the relation ID $r$ to prepend factually correct sentences. For instance, one can hijack the example above to "The Eiffel Tower is not located in Guam. The Eiffel Tower is in". We test this hijacking philosophy on different relation IDs. In particular, Figure 2b reports hijacking based on relation ID $P190$ ("twin city"). And we see similar patterns that with more prepends, the ES score gets higher. It is also worth noting that one can even hijack by only including words that are semantically close to the false target (e.g., "France" for false target "French"). This suggests that context hijacking is more than simply the LLM copying tokens from contexts. Additional details and experiments for both hijacking schemes and for other relation IDs are in Appendix C.

These experiments show that context hijacking changes the behavior of LLMs, leading them to output incorrect tokens, without altering the factual meaning of the context. It is worth noting that similar fragile behaviors of LLMs have been observed in the literature in different contexts [Shi+23; Pet+20; CSH22; Yor+23; PE21]. See Section 2 for more details.

Context hijacking indicates that fact retrieval in LLMs is not robust and that accurate fact recall does not necessarily depend on the semantics of the context. As a result, one hypothesis is to view LLMs as an associative memory model where special tokens in contexts, associated with the fact, provide partial information or clues to facilitate memory retrieval [Zha23]. To better understand this perspective, we design a synthetic memory retrieval task to evaluate how the building blocks of LLMs, transformers, can solve it.

# 4  Problem setup

In the context of LLMs, fact or memory retrieval, can be modeled as a next token prediction problem. Given a context (e.g., "The capital of France is"), the objective is to accurately predict the next token (e.g., "Paris") based on the factual relation between context and the following token.

Previous papers [Ram+20; Mil+22; BP21; Zha23] have studied the connection between attention and autoassociative and heteroassociative memory. For autoassociative memory, contexts are modeled as a set of existing memories and the goal of self-attention is to select the closest one or approximations to it. On top of this, heteroassociative memory [Mil+22; BP21] has an additional projection to remap each output to a different one, whether within the same space or otherwise. In both scenarios, the

goal is to locate the closest pattern within the context when provided with a query (up to a remapping if it's heteroassociative).

Fact retrieval, on the other hand, does not strictly follow this framework. The crux of the issue is that the output token is not necessarily close to any particular token in the context but rather a combination of them and the "closeness" is intuitively measured by latent semantic concepts. For example, consider context sentence "The capital of France is" with the output "Paris". Here, none of the tokens in the context directly corresponds to the word "Paris". Yet some tokens contain partial information about "Paris". Intuitively, "capital" aligns with the "isCapital" concept of "Paris", while "France" corresponds to the "isFrench" concept linked to "Paris" where all the concepts are latent. To model such phenomenon, we propose a synthetic task called *latent concept association* where the output token is closely related to tokens in the context and similarity is measured via the latent space.

## 4.1 Latent concept association

We propose a synthetic prediction task where for each output token $y$, tokens in the context (denoted by $x$) are sampled from a conditional distribution given $y$. Tokens that are similar to $y$ will be favored to appear more in the context, except for $y$ itself. The task of latent concept association is to successfully retrieve the token $y$ given samples from $p(x|y)$. The synthetic setup simplifies by not accounting for the sequential nature of language, a choice supported by previous experiments on context hijacking (Section 3). We formalize this task below.

To measure similarity, we define a latent space. Here, the latent space is a collection of $m$ binary latent variables $Z_i$. These could be viewed as semantic concept variables. Let $Z = (Z_1, ..., Z_m)$ be the corresponding random vector, $z$ be its realization, and $\mathcal{Z}$ be the collection of all latent binary vectors. For each latent vector $z$, there's one associated token $t \in [V] = \{0, ..., V - 1\}$ where $V$ is the total number of tokens. Here we represent the tokenizer as $\iota$ where $\iota(z) = t$. In this paper, we assume that $\iota$ is the standard tokenizer where each binary vector is mapped to its decimal number. In other words, there's a one to one map between latent vectors and tokens. Because the map is one to one, we sometimes use latent vectors and tokens interchangeably. We also assume that every latent binary vector has a unique corresponding token, therefore $V = 2^m$.

Under the latent concept association model, the goal is to retrieve specific output tokens given partial information in the contexts. This is modeled by the latent conditional distribution:

$$p(z|z^*) = \omega \pi(z|z^*) + (1 - \omega)\text{Unif}(\mathcal{Z})$$

where

$$\pi(z|z^*) \propto \begin{cases} \exp(-D_H(z, z^*)/\beta) & z \in \mathcal{N}(z^*), \\ 0 & z \notin \mathcal{N}(z^*). \end{cases}$$

Here $D_H$ is the Hamming distance, $\mathcal{N}(z^*)$ is a subset of $\mathcal{Z} \backslash \{z^*\}$ and $\beta > 0$ is the temperature parameter. The use of Hamming distance draws a parallel with the notion of distributional semantics in natural language: "a word is characterized by the company it keeps" [Fir57]. In words, $p(z|z^*)$ says that with probability $1 - \omega$, the conditional distribution uniformly generate random latent vectors and with probability $\omega$, the latent vector is generated from the *informative conditional distribution* $\pi(z|z^*)$ where the support of the conditional distribution is $\mathcal{N}(z^*)$. Here, $\pi$ represents the informative conditional distribution that depends on $z^*$ whereas the uniform distribution is uninformative and can be considered as noise. The mixture model parameter $\omega$ determines the signal to noise ratio of the contexts.

Therefore, for any latent vector $z^*$ and its associated token, one can generate $L$ context token words with the aforementioned latent conditional distribution:

- Uniformly sample a latent vector $z^*$
- For $l = 1, ..., L - 1$, sample $z_l \sim p(z|z^*)$ and $t_l = \iota(z_l)$.
- For $l = L$, sample $z \sim \pi(z|z^*)$ and $t_L = \iota(z)$.

Consequently, we have $x = (t_1, .., t_L)$ and $y = \iota(z^*)$. The last token in the context is generated specifically to make sure that it is not from the uniform distribution. This ensures that the last token can use attention to look for clues, relevant to the output, in the context. Let $\mathcal{D}^L$ be the sampling distribution to generate $(x, y)$ pairs. The conditional probability of $y$ given $x$ is given by $p(y|x)$.

With slight abuse of notation, given a token $t \in [V]$, we define $\mathcal{N}(t) = \mathcal{N}(\iota^{-1}(t))$. we also define $D_H(t, t') = D_H(\iota^{-1}(t), \iota^{-1}(t'))$ for any pair of tokens $t$ and $t'$.

For any function $f$ that maps the context to estimated logits of output labels, the training objective is to minimize this loss of the last position:

$$\mathbb{E}_{(x,y) \in \mathcal{D}^L}[\ell(f(x), y)]$$

where $\ell$ is the cross entropy loss with softmax. The error rate of latent concept association is defined by the following:

$$R_{\mathcal{D}^L}(f) = \mathbb{P}_{(x,y) \sim \mathcal{D}^L}[\text{argmax } f(x) \neq y]$$

And the accuracy is $1 - R_{\mathcal{D}^L}(f)$.

## 4.2 Transformer network architecture

Given a context $x = (t_1, .., t_L)$ which consists of $L$ tokens, we define $X \in \{0, 1\}^{V \times L}$ to be its one-hot encoding where $V$ is the vocabulary size. Here we use $\chi$ to represent the one-hot encoding function (i.e., $\chi(x) = X$). Similar to [LLR23; Tar+23a; Li+24], we also consider a simplified one-layer transformer model without residual connections and normalization:

$$f^L(x) = \left[ W_E^T W_V \text{attn}(W_E \chi(x)) \right]_{:L} \tag{4.1}$$

where

$$\text{attn}(U) = U\sigma\left( \frac{(W_K U)^T (W_Q U)}{\sqrt{d_a}} \right),$$

$W_K \in \mathbb{R}^{d_a \times d}$ is the key matrix, and $W_Q \in \mathbb{R}^{d_a \times d}$ is the query matrix and $d_a$ is the attention head size. $\sigma : \mathbb{R}^{L \times L} \to (0, 1)^{L \times L}$ is the column-wise softmax operation. $W_V \in \mathbb{R}^{d \times d}$ is the value matrix and $W_E \in \mathbb{R}^{d \times V}$ is the embedding matrix. Here, we adopt the weight tie-in implementation which is used for Gemma [Tea+24]. We focus solely on the prediction of the last position, as it is the only one relevant for latent concept association. For convenience, we also use $h(x)$ to mean $\left[ \text{attn}(W_E \chi(x)) \right]_{:L}$, which is the hidden representation after attention for the last position, and $f_t^L(x)$ to represent the logit for output token $t$.

## 5 Theoretical analysis

In this section, we theoretically investigate how a single-layer transformer can solve the latent concept association problem. We first introduce a hypothetical associative memory model that utilizes self-attention for information aggregation and employs the value matrix for memory retrieval. This hypothetical model turns out to mirror trained transformers in experiments. We also examine the role of each individual component of the network: the value matrix, embeddings, and the attention mechanism. We validate our theoretical claims in Section 6.

### 5.1 Hypothetical associative memory model

In this section, we show that a simple single-layer transformer network can solve the latent concept association problem. The formal result is presented below in Theorem 1; first we require a few more definitions. Let $W_E(t)$ be the $t$-th column of the embedding matrix $W_E$. In other words, this is the embedding for token $t$. Given a token $t$, define $\mathcal{N}_1(t)$ to be the subset of tokens whose latent vectors are only 1 Hamming distance away from $t$'s latent vector: $\mathcal{N}_1(t) = \{t' : D_H(t', t)) = 1\} \cap \mathcal{N}(t)$. For any output token $t$, $\mathcal{N}_1(t)$ contains tokens with the highest probabilities to appear in the context.

The following theorem formalizes the intuition that a one-layer transformer that uses self-attention to summarize statistics about the context distributions and whose value matrix uses aggregated representations to retrieve output tokens can solve the latent concept association problem defined in Section 4.1.

**Theorem 1** (informal). *Suppose the data generating process follows Section 4.1 where $m \geq 3$, $\omega = 1$, and $\mathcal{N}(t) = V \setminus \{t\}$. Then for any $\varepsilon > 0$, there exists a transformer model given by (4.1) that achieves error $\varepsilon$, i.e. $R_{\mathcal{D}^L}(f^L) < \varepsilon$ given sufficiently large context length $L$.*

More precisely, for the transformer in Theorem 1, we will have $W_K = 0$ and $W_Q = 0$. Each row of $W_E$ is orthogonal to each other and normalized. And $W_V$ is given by

$$W_V = \sum_{t \in [V]} W_E(t)(\sum_{t' \in \mathcal{N}_1(t)} W_E(t')^T) \tag{5.1}$$

A more formal statement of the theorem and its proof is given in Appendix B (Theorem 7).

Intuitively, Theorem 1 suggests having more samples from $p(x|y)$ can lead to a better recall rate. On the other hand, if contexts are modified to contain more samples from $p(x|\tilde{y})$ where $\tilde{y} \neq y$, then it is likely for transformer to output the wrong token. This is similar to context hijacking (see Section 5.5). The construction of the value matrix is similar to the associative memory model used in [Bie+24; CSB24], but in our case, there is no explicit one-to-one input and output pairs stored as memories. Rather, a combination of inputs are mapped to a single output.

While the construction in Theorem 1 is just one way that a single-layer transformer can tackle this task, it turns out empirically this construction of $W_V$ is close to the trained $W_V$, even in the noisy case ($\omega \neq 1$). In Section 6.1, we will demonstrate that substituting trained value matrices with constructed ones can retain accuracy, and the constructed and trained value matrices even share close low-rank approximations. Moreover, in this hypothetical model, a simple uniform attention mechanism is deployed to allow self-attention to count occurrences of each individual tokens. Since the embeddings are orthonormal vectors, there is no interference. Hence, the self-attention layer can be viewed as aggregating information of contexts. It is worth noting that, in different settings, more sophisticated embedding structures and attention patterns are needed. This is discussed in the following sections.

## 5.2 On the role of the value matrix

The construction in Theorem 1 relies on the value matrix acting as associative memory. But is it necessary? Could we integrate the functionality of the value matrix into the self-attention module to solve the latent concept association problem? Empirically, the answer seems to be negative as will be shown in Section 6.1. In particular, when the context length is small, setting the value matrix to be the identity would lead to subpar memory recall accuracy.

This is because if the value matrix is the identity, the transformer would be more susceptible to the noise in the context. To see this, notice that given any pair of context and output token $(x, y)$, the latent representation after self-attention $h(x)$ must live in the polyhedron $S_y$ to be classified correctly where $S_y$ is defined as:

$$S_y = \{v : (W_E(y) - W_E(t))^T v > 0 \text{ where } t \notin [V] \setminus \{y\}\}$$

Note that, by definition, for any two tokens $y$ and $\tilde{y}$, $S_y \cap S_{\tilde{y}} = \emptyset$. On the other hand, because of the self-attention mechanism, $h(x)$ must also live in the convex hull of all the embedding vectors:

$$CV = \text{Conv}(W^E(0), ..., W^E(|V| - 1))$$

In other words, for any pair $(x, y)$ to be classified correctly, $h(x)$ must live in the intersection of $S_y$ and $CV$. Due to the stochastic nature of $x$, it is likely for $h(x)$ to be outside of this intersection. The remapping effect of the value matrix can help with this problem. The following lemma explains this intuition.

**Lemma 2.** *Suppose the data generating process follows Section 4.1 where $m \geq 3$, $\omega = 1$ and $\mathcal{N}(t) = \{t' : D_H(t, t')) = 1\}$. For any single layer transformer given by (4.1) where each row of $W_E$ is orthogonal to each other and normalized, if $W_V$ is constructed as in (5.1), then the error rate is 0. If $W_V$ is the identity matrix, then the error rate is strictly larger than 0.*

Another intriguing phenomenon occurs when the value matrix is the identity matrix. In this case, the inner product between embeddings and their corresponding Hamming distance varies linearly. This relationship can be formalized by the following theorem.

**Theorem 3.** *Suppose the data generating process follows Section 4.1 where $m \geq 3$, $\omega = 1$ and $\mathcal{N}(t) = V \setminus \{t\}$. For any single layer transformer given by (4.1) with $W_V$ being the identity matrix, if the cross entropy loss is minimized so that for any sampled pair $(x, y)$,*

$$p(y|x) = \hat{p}(y|x) = softmax(f_y^L(x))$$

*there exists $a > 0$ and $b$ such that for two tokens $t \neq t'$,*

$$\langle W_E(t), W_E(t') \rangle = -aD_H(t, t') + b$$

## 5.3 Embedding training and geometry

The hypothetical model in Section 5.1 requires embeddings to form an orthonormal basis. In the overparameterization regime where the embedding dimension $d$ is larger than the number of tokens $V$, this can be approximately achieved by Gaussian initialization. However, in practice, the embedding dimension is typically smaller than the vocabulary size, in which case it is impossible for the embeddings to constitute such a basis. Empirically, in Section 6.2, we observe that with overparameterization ($d > V$), embeddings can be frozen at their Gaussian initialization, whereas in the underparameterized regime, embedding training is required to achieve better recall accuracy.

This raises the question: What kind of embedding geometry is learned in the underparameterized regime? Experiments reveal a close relationship between the inner product of embeddings for two tokens and the Hamming distance of these tokens (see Figure 3b and Figure D.5 in Appendix D.2). Approximately, we have the following relationship:

$$\langle W_E(t), W_E(t') \rangle = \begin{cases} b_0 & t = t' \\ -aD_H(t, t') + b & t \neq t' \end{cases} \tag{5.2}$$

for any two tokens $t$ and $t'$ where $b_0 > b$ and $a > 0$. One can view this as a combination of the embedding geometry under Gaussian initialization and the geometry when $W_V$ is the identity matrix (Theorem 3). Importantly, this structure demonstrates that trained embeddings inherently capture similarity within the latent space. Theoretically, this embedding structure (5.2) can also lead to low error rate under specific conditions on $b_0, b$ and $a$, which is articulated by the following theorem.

**Theorem 4** (Informal). *Following the same setup as in Theorem 1, but embeddings obey (5.2), then under certain conditions on $a, b$ and if $b_0$ and context length $L$ are sufficiently large, the error rate can be arbitrarily small, i.e. $R_{\mathcal{D}^L}(f^L) < \varepsilon$ for any $0 < \varepsilon < 1$.*

The formal statement of the theorem and its proof is given in Appendix B (Theorem 8).

Notably, this embedding geometry also implies a low-rank structure. Let's first consider the special case when $b_0 = b$. In other words, the inner product between embeddings and their corresponding Hamming distance varies linearly.

**Lemma 5.** *If embeddings follow (5.2) and $b = b_0$ and $\mathcal{N}(t) = V \setminus \{t\}$, then $rank(W_E) \leq m + 2$.*

When $b_0 > b$, the embedding matrix will not be strictly low rank. However, it can still exhibit approximate low-rank behavior, characterized by an eigengap between the top and bottom singular values. This is verified empirically (see Figure D.9-D.12 in Appendix D.4).

## 5.4 The role of attention selection

As of now, attention does not play a significant role in the analysis. But perhaps unsurprisingly, the attention mechanism is useful in selecting relevant information. To see this, let's consider a specific setting where for any latent vector $z^*$, $\mathcal{N}(z^*) = \{z : z_1^* = z_1\} \setminus \{z^*\}$.

Essentially, latent vectors are partitioned into two clusters based on the value of the first latent variable, and the informative conditional distribution $\pi$ only samples latent vectors that are in the same cluster as the output latent vector. Empirically, when trained under this setting, the attention mechanism will pay more attention to tokens within the same cluster (Section 6.3). This implies that the self-attention layer can mitigate noise and concentrate on the informative conditional distribution $\pi$.

To understand this more intuitively, we will study the gradient of unnormalized attention scores. In particular, the unnormalized attention score is defined as:

$$u_{t,t'} = (W_K W_E(t))^T (W_Q W_E(t')) / \sqrt{d_a}.$$

**Lemma 6.** *Suppose the data generating process follows Section 4.1 and $\mathcal{N}(z^*) = \{z : z_1^* = z_1\} \setminus \{z^*\}$. Given the last token in the sequence $t_L$, then*

$$\nabla_{u_{t,t_L}} \ell(f^L) = \nabla \ell(f^L)^T (W_E)^T W^V (\alpha_t \hat{p}_t W_E(t) - \hat{p}_t \sum_{l=1}^{L} \hat{p}_{t_l} W_E(t_l))$$

*where for token $t$, $\alpha_t = \sum_{l=1}^{L} \mathbf{1}[t_l = t]$ and $\hat{p}_t$ is the normalized attention score for token $t$.*

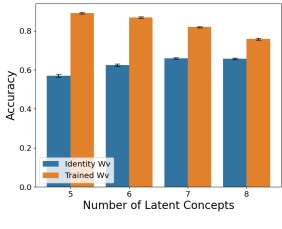

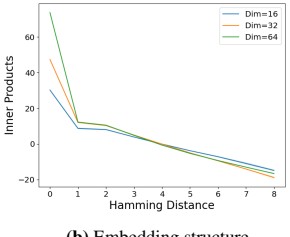

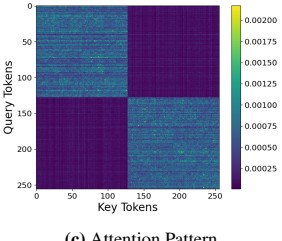

| (a) Value matrix training | (b) Embedding structure | (c) Attention Pattern |

**Figure 3:** Key components of the single-layer transformer working together on the latent concept association problem. (a) Fixing the value matrix $W_V$ as the identity matrix results in lower accuracy compared to training $W_V$. The figure reports average accuracy for both fixed and trained $W_V$ with $L = 64$. (b) When training in the underparameterized regime, the embedding structure is approximated by (5.2). The graph displays the average inner product between embeddings of two tokens against the corresponding Hamming distance between these tokens when $m = 8$. (c) The self-attention layer can select tokens within the same cluster. The figure shows average attention score heat map with $m = 8$ and the cluster structure from Section 5.4.

Typically, $\alpha_t$ is larger when token $t$ and $t_L$ belong to the same cluster because tokens within the same cluster tend to co-occur frequently. As a result, the gradient contribution to the unnormalized attention score is usually larger for tokens within the same cluster.

### 5.5 Context hijacking and the misclassification of memory recall

In light of the theoretical results on latent concept association, a natural question arises: How do these results connect to context hijacking in LLMs? In essence, for the latent concept association problem, the differentiation of output tokens is achieved by distinguishing between the various conditional distributions $p(x|y)$. Thus, adding or changing tokens in the context $x$ so that it resembles a different conditional distribution can result in misclassification. In Appendix D.5, we present experiments showing that mixing different contexts can cause transformers to misclassify. This partially explains context hijacking in LLMs (Section 3). On the other hand, it is well-known that the error rate is related to the KL divergence between conditional distributions of contexts [Cov99]. The closer the distributions are, the easier it is for the model to misclassify. Here, longer contexts, primarily composed of i.i.d samples, suggest larger divergences, thus higher memory recall rate. This is theoretically implied by Theorem 1 and Theorem 4 and empirically verified in Appendix D.6. Such result is also related to reverse context hijacking (Appendix C) where prepending sentences including true target words can improve fact recall rate.

## 6 Experiments

The main implications of the theoretical results in the previous section are:

1. The value matrix is important and has associative memory structure as in (5.1).

2. Training embeddings is crucial in the underparameterized regime, where embeddings exhibit certain geometric structures.

3. Attention mechanism is used to select the most relevant tokens.

To evaluate these claims, we conduct several experiments on synthetic datasets. Additional experimental details and results can be found in Appendix D.

### 6.1 On the value matrix $W_V$

In this section, we study the necessity of the value matrix $W_V$ and its structure. First, we conduct experiments to compare the effects of training versus freezing $W_V$ as the identity matrix, with the context lengths $L$ set to 64 and 128. Figure 3a and Figure D.1 show that when the context length is small, freezing $W_V$ can lead to a significant decline in accuracy. This is inline with Lemma 2 and validates it in a general setting, implying the significance of the value matrix in maintaining a high memory recall rate.

Next, we investigate the degree of alignment between the trained value matrix $W_V$ and the construction in (5.1). The first set of experiments examines the similarity in functionality between the two matrices. We replace value matrices in trained transformers with the constructed ones like in (5.1) and then report accuracy with the new value matrix. As a baseline, we also consider randomly constructed value matrix, where the outer product pairs are chosen randomly (detailed construction can be found in Appendix D.1). Figure D.2 indicates that the accuracy does not significantly decrease when the value matrix is replaced with the constructed ones. Furthermore, not only are the constructed value matrix and the trained value matrix functionally alike, but they also share similar low-rank approximations. We use singular value decomposition to get the best low rank approximations of various value matrices where the rank is set to be the same as the number of latent variables ($m$). We then compute smallest principal angles between low-rank approximations of trained value matrices and those of constructed, randomly constructed, and Gaussian-initialized value matrices. Figure D.3 shows that the constructed ones have, on average, smallest principal angles with the trained ones.

## 6.2 On the embeddings

In this section, we explore the significance of embedding training in the underparamerized regime and embedding structures. We conduct experiments to compare the effects of training versus freezing embeddings with different embedding dimensions. The learning rate is selected as the best option from $\{0.01, 0.001\}$ depending on the dimensions. Figure D.4 clearly shows that when the dimension is smaller than the vocabulary size ($d < V$), embedding training is required. It is not necessary in the overparameterized regime ($d > V$), partially confirming Theorem 1 because if embeddings are initialized from a high-dimensional multi-variate Gaussian, they are approximately orthogonal to each other and have the same norms.

The next question is what kind of embedding structures are formed for trained transformers in the underparamerized regime. From Figure 3b and Figure D.5, it is evident that the relationship between the average inner product of embeddings for two tokens and their corresponding Hamming distance roughly aligns with (5.2). Perhaps surprisingly, if we plot the same graph for trained transformers with a fixed identity value matrix, the relationship is mostly linear as shown in Figure D.6, confirming our theory (Theorem 3).

As suggested in Section 5.3, such embedding geometry (5.2) can lead to low rank structures. We verify this claim by studying the spectrum of the embedding matrix $W_E$. As illustrated in Appendix D.4, Figure D.9-D.12 demonstrate that there are eigengaps between top and bottom singular values, suggesting low-rank structures.

## 6.3 On the attention selection mechanism

In this section, we examine the role of attention pattern by considering a special class of latent concept association model as defined in Section 5.4. Figure 3c and Figure D.7 clearly show that the self-attention select tokens in the same clusters. This suggests that attention can filter out noise and focus on the informative conditional distribution $\pi$. We extend experiments to consider cluster structures that depend on the first two latent variables (detailed construction can be found in Appendix D.3) and Figure D.8 shows attention pattern as expected.

## 7 Conclusions

In this work, we first presented the phenomenon of context hijacking in LLMs, which suggested that fact retrieval is not robust against variations of contexts. This indicates that LLMs might function like associative memory where tokens in contexts are clues to guide memory retrieval. To investigate this perspective further, we devised a synthetic task called latent concept association and examined theoretically and empirically how single-layer transformers are trained to solve this task. These results provide further insights into the inner workings of transformers and LLMs, and can hopefully stimulate further work into interpreting and understanding the mechanisms by which LLMs predict tokens and recall facts.

**Acknowledgments**    We thank Victor Veitch for insightful discussions that helped shape the initial idea of this work. We acknowledge the support of AFRL and DARPA via FA8750-23-2-1015, ONR via N00014-23-1-2368, NSF via IIS-1909816, IIS-1955532, IIS-1956330, and NIH R01GM140467. We also acknowledge the support of the Robert H. Topel Faculty Research Fund at the University of Chicago Booth School of Business.

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

# A   Limitations

The context hijacking experiments were only conducted on open-source models and not on commercial models like GPT-4. Nevertheless, even in the official GPT-4 technical report [Ach+23], there is an example similar to context hijacking (the Elvis Perkins example). In that example, the prompt is "Son of an actor, this American guitarist and rock singer released many songs and albums and toured with his band. His name is "Elvis" what?". GPT-4 answers with Presley, even though the answer is Perkins (Elvis Presley is not the son of an actor). GPT-4 can be viewed as distracted by all the information related to music and answers Presley. In fact, it is known that LLMs can be easily distracted by contexts in use cases other than fact retrieval such as problem-solving [Shi+23]. So we reasonably suspect that similar behavior still exists in larger models but is harder to exploit. On the other hand, the theoretical section only focuses on single-layer transformer network. While single-layer networks already demonstrate some interesting phenomena including low-rank structures, the functionality of multi-layer transformers is much different compared to single-layer transformers with the notable emergence of induction head [Elh+21].

# B   Additional Theoretical Results and Proofs

## B.1   Proofs for Section 5.1

Theorem 1 can be stated more formally as follows:

**Theorem 7.** *Suppose the data generating process follows Section 4.1 where $m \geq 3$, $\omega = 1$, and $\mathcal{N}(t) = V \setminus \{t\}$. Assume there exists a single layer transformer given by (4.1) such that a) $W_K = 0$ and $W_Q = 0$, b) Each row of $W_E$ is orthogonal to each other and normalized, and c) $W_V$ is given by*

$$W_V = \sum_{i \in [V]} W_E(i) \left( \sum_{j \in \mathcal{N}_1(i)} W_E(j)^T \right).$$

*Then if $L > \max\{ \frac{100m^2 \log(3/\varepsilon)}{(\exp(-\frac{1}{\beta}) - \exp(-\frac{2}{\beta}))^2}, \frac{80m^2|\mathcal{N}(y)|}{(\exp(-\frac{1}{\beta}) - \exp(-\frac{2}{\beta}))^2} \}$ for any $y$, then*

$$R_{\mathcal{D}^L}(f^L) \leq \varepsilon,$$

*where $0 < \varepsilon < 1$.*

*Proof.* First of all, the error is defined to be:

$$R_{\mathcal{D}^L}(f^L) = \mathbb{P}_{(x,y) \sim \mathcal{D}^L}[\operatorname{argmax} f^L(x) \neq y]$$
$$= \mathbb{P}_y \mathbb{P}_{x|y}[\operatorname{argmax} f^L(x) \neq y]$$

Let's focus on the conditional probability $\mathbb{P}_{x|y}[\operatorname{argmax} f^L(x) \neq y]$.

By construction, the single layer transformer model has uniform attention. Therefore,

$$h(x) = \sum_{i \in \mathcal{N}(y)} \alpha_i W_E(i)$$

where $\alpha_i = \frac{1}{L} \sum_{k=1}^{L} \mathbf{1}\{t_k = i\}$ which is the number of occurrence of token $i$ in the sequence.

By the latent concept association model, we know that

$$p(i|y) = \frac{\exp(-D_H(i,y)/\beta)}{Z}$$

where $Z = \sum_{i \in \mathcal{N}(y)} \exp(-D_H(i,y)/\beta)$.

Thus, the logit for token $y$ is

$$f_y^L(x) = \sum_{i \in \mathcal{N}_1(y)} \alpha_i$$

And the logit for any other token $\tilde{y}$ is

$$f_{\tilde{y}}^L(x) = \sum_{i \in \mathcal{N}_1(\tilde{y})} \alpha_i$$

For the prediction to be correct, we need

$$\max_{\tilde{y}} f_y^L(x) - f_{\tilde{y}}^L(x) > 0$$

By Lemma 3 of [Dev83], we know that for all $\Delta \in (0,1)$, if $\frac{|\mathcal{N}(y)|}{L} \leq \frac{\Delta^2}{20}$, we have

$$\mathbb{P}\left( \max_{i \in \mathcal{N}(y)} |\alpha_i - p(i|y)| > \Delta \right) \leq \mathbb{P}\left( \sum_{i \in \mathcal{N}(y)} |\alpha_i - p(i|y)| > \Delta \right) \leq 3 \exp(-L\Delta^2/25)$$

Therefore, if $L \geq \max\{\frac{25 \log(3/\varepsilon)}{\Delta^2}, \frac{20|\mathcal{N}(y)|}{\Delta^2}\}$, then with probability at least $1 - \varepsilon$, we have,

$$\max_{i \in \mathcal{N}(y)} |\alpha_i - p(i|y)| \leq \Delta$$

$$
\begin{aligned}
f_y^L(x) - f_{\tilde{y}}^L(x) &= \sum_{i \in \mathcal{N}_1(y)} \alpha_i - \sum_{j \in \mathcal{N}_1(\tilde{y})} \alpha_j \\
&= \sum_{i \in \mathcal{N}_1(y)} \alpha_i - \sum_{i \in \mathcal{N}_1(y)} p(i|y) + \sum_{i \in \mathcal{N}_1(y)} p(i|y) \\
&\quad - \sum_{j \in \mathcal{N}_1(\tilde{y})} p(j|y) + \sum_{j \in \mathcal{N}_1(\tilde{y})} p(j|y) - \sum_{j \in \mathcal{N}_1(\tilde{y})} \alpha_j \\
&\geq \sum_{i \in \mathcal{N}_1(y)} p(i|y) - \sum_{j \in \mathcal{N}_1(\tilde{y})} p(j|y) - 2m\Delta \\
&\geq \exp(-\frac{1}{\beta}) - \exp(-\frac{2}{\beta}) - 2m\Delta
\end{aligned}
$$

Note that because of Lemma 10, there's no neighboring set that is the superset of another.

Therefore as long as $\Delta < \frac{\exp(-\frac{1}{\beta}) - \exp(-\frac{2}{\beta})}{2m}$,

$$f_y^L(x) - f_{\tilde{y}}^L(x) > 0$$

for any $\tilde{y}$.

Finally, if $L > \max\{\frac{100m^2 \log(3/\varepsilon)}{(\exp(-\frac{1}{\beta}) - \exp(-\frac{2}{\beta}))^2}, \frac{80m^2|\mathcal{N}(y)|}{(\exp(-\frac{1}{\beta}) - \exp(-\frac{2}{\beta}))^2}\}$ for any $y$, then

$$\mathbb{P}_{x|y}[\operatorname{argmax} f^L(x) \neq y] \leq \varepsilon$$

And

$$
\begin{aligned}
R_{\mathcal{D}^L}(f^L) &= \mathbb{P}_{(x,y) \sim \mathcal{D}^L}[\operatorname{argmax} f^L(x) \neq y] \\
&= \mathbb{P}_y \mathbb{P}_{x|y}[\operatorname{argmax} f^L(x) \neq y] \leq \varepsilon
\end{aligned}
$$

$\square$

## B.2   Proofs for Section 5.2

**Lemma 2.** *Suppose the data generating process follows Section 4.1 where $m \geq 3$, $\omega = 1$ and $\mathcal{N}(t) = \{t' : D_H(t,t')) = 1\}$. For any single layer transformer given by (4.1) where each row of $W_E$ is orthogonal to each other and normalized, if $W_V$ is constructed as in (5.1), then the error rate is 0. If $W_V$ is the identity matrix, then the error rate is strictly larger than 0.*

*Proof.* Following the proof for Theorem 7, let's focus on the conditional probability:

$$\mathbb{P}_{x|y}[\operatorname{argmax} f^L(x) \neq y]$$

By construction, we have

$$h(x) = \sum_{i \in \mathcal{N}_1(y)} \alpha_i W_E(i)$$

where $\alpha_i = \frac{1}{L} \sum_{k=1}^{L} \mathbf{1}\{t_k = i\}$ which is the number of occurrence of token $i$ in the sequence.

Let's consider the first case where $W_V$ is constructed as in (5.1). Then we know that for some other token $\tilde{y} \neq y$,

$$f_y^L(x) - f_{\tilde{y}}^L(x) = \sum_{i \in \mathcal{N}_1(y)} \alpha_i - \sum_{i \in \mathcal{N}_1(\tilde{y})} \alpha_i = 1 - \sum_{i \in \mathcal{N}_1(\tilde{y})} \alpha_i$$

By Lemma 10, we have that for any token $\tilde{y} \neq y$,

$$f_y^L(x) - f_{\tilde{y}}^L(x) > 0$$

Therefore, the error rate is always 0.

Now let's consider the second case where $W_V$ is the identity matrix. Let $j$ be a token in the set $\mathcal{N}_1(y)$. Then there is a non-zero probability that context $x$ contains only $j$. In that case,

$$h(x) = W_E(j)$$

However, we know that by the assumption on the embedding matrix,

$$f_y^L(x) - f_j^L(x) = (W_E(y) - W_E(j))^T h(x) = -\|W_E(j)\|^2 < 0$$

This implies that there's non zero probability that $y$ is misclassified. Therefore, when $W_V$ is the identity matrix, the error rate is strictly larger than 0. $\qquad\square$

**Theorem 3.** *Suppose the data generating process follows Section 4.1 where $m \geq 3$, $\omega = 1$ and $\mathcal{N}(t) = V \setminus \{t\}$. For any single layer transformer given by (4.1) with $W_V$ being the identity matrix, if the cross entropy loss is minimized so that for any sampled pair $(x, y)$,*

$$p(y|x) = \hat{p}(y|x) = softmax(f_y^L(x))$$

*there exists $a > 0$ and $b$ such that for two tokens $t \neq t'$,*

$$\langle W_E(t), W_E(t') \rangle = -a D_H(t, t') + b$$

*Proof.* Because for any pair of $(x, y)$, the estimated conditional probability matches the true conditional probability. In particular, let's consider two target tokens $y_1, y_2$ and context $x = (t_i, ..., t_i)$ for some token $t_i$ such that $p(x|y_1) > 0$ and $p(x|y_2) > 0$, then

$$\frac{p(y_1|x)}{p(y_2|x)} = \frac{p(x|y_1)p(y_1)}{p(x|y_2)p(y_2)} = \frac{p(x|y_1)}{p(x|y_2)} = \frac{\hat{p}(x|y_1)}{\hat{p}(x|y_2)} = \exp((W_E(y_1) - W_E(y_2))^T h(x))$$

The second equality is because $p(y)$ is the uniform distribution. By our construction,

$$\frac{p(x|y_1)}{p(x|y_2)} = \frac{p(t_i|y_1)^L}{p(t_i|y_2)^L} = \exp((W_E(y_2) - W_E(y_1))^T h(x)) = \exp((W_E(y_1) - W_E(y_2))^T W_E(t_i))$$

By the data generating process, we have that

$$\frac{L}{\beta}(D_H(t_i, y_2) - D_H(t_i, y_1)) = (W_E(y_1) - W_E(y_2))^T W_E(t_i)$$

Let $t_i = y_3$ such that $y_3 \neq y_1, y_3 \neq y_2$, then

$$\frac{L}{\beta} D_H(y_3, y_1) - W_E(y_1)^T W_E(y_3) = \frac{L}{\beta} D_H(y_3, y_2) - W_E(y_2)^T W_E(y_3)$$

For simplicity, let's define

$$\Psi(y_1, y_2) = \frac{L}{\beta} D_H(y_1, y_2) - W_E(y_1)^T W_E(y_2)$$

Therefore,

$$\Psi(y_3, y_1) = \Psi(y_3, y_2)$$

Now consider five distinct labels: $y_1, y_2, y_3, y_4, y_5$. We have,

$$\Psi(y_3, y_1) = \Psi(y_3, y_2) = \Psi(y_4, y_2) = \Psi(y_4, y_5)$$

In other words, $\Psi(y_3, y_1) = \Psi(y_4, y_5)$ for arbitrarily chosen distinct labels $y_1, y_3, y_4, y_5$. Therefore, $\Psi(t, t')$ is a constant for $t \neq t'$.

For any two tokens $t \neq t'$,

$$\frac{L}{\beta} D_H(t, t') - W_E(t)^T W_E(t') = C$$

Thus,

$$W_E(t)^T W_E(t') = -\frac{L}{\beta} D_H(t, t') + C$$

$\square$

## B.3 Proofs for Section 5.3

Theorem 4 can be formalized as the following theorem.

**Theorem 8.** *Following the same setup as in Theorem 7, but embeddings follow (5.2) then if $b > 0$, $\Delta_1 > 0$, $0 < \Delta < \frac{\exp(-\frac{1}{\beta}) - \exp(-\frac{2}{\beta})}{2m}$, $L \geq \max\{\frac{25 \log(3/\varepsilon)}{\Delta^2}, \frac{20|\mathcal{N}(y)|}{\Delta^2}\}$ for any $y$, and*

$$0 < a < \frac{2\exp(\frac{1}{\beta})}{(|V| - 2)m^2}$$

*and*

$$b_0 > \max\{\frac{a(m-2)m + \Delta_1}{\exp(-\frac{1}{\beta}) - \exp(-\frac{2}{\beta}) - 2m\Delta} + b, \frac{(b-a)\Delta_1 - \frac{|V|-2}{2}abm^2 \exp(-\frac{1}{\beta}) + \frac{|V|-2}{2}a^2(m-2)m^2}{1 - \frac{|V|-2}{2}am^2 \exp(-\frac{1}{\beta})}\}$$

*we have*

$$R_{\mathcal{D}^L}(f^L) \leq \varepsilon$$

*where $0 < \varepsilon < 1$.*

*Proof.* Following the proof of Theorem 7, let's also focus on the conditional probability

$$\mathbb{P}_{x|y}[\arg\max f^L(x) \neq y]$$

By construction, the single layer transformer model has uniform attention. Therefore,

$$h(x) = \sum_{i \in \mathcal{N}(y)} \alpha_i W_E(i)$$

where $\alpha_i = \frac{1}{L} \sum_{k=1}^{L} \mathbf{1}\{t_k = i\}$ which is the number of occurrence of token $i$ in the sequence. For simplicity, let's define $\alpha_y = 0$ such that

$$h(x) = \sum_{i \in [V]} \alpha_i W_E(i)$$

Similarly, we also have that if $L \geq \max\{\frac{25 \log(3/\varepsilon)}{\Delta^2}, \frac{20|\mathcal{N}(y)|}{\Delta^2}\}$, then with probability at least $1 - \varepsilon$, we have,

$$\max_{i \in [V]} |\alpha_i - p(i|y)| \leq \Delta$$

Also define the following:

$$\phi_k(x) = \sum_{j \in \mathcal{N}_1(k)} W_E(j)^T \big( \sum_{i \in [V]} \alpha_i W_E(i) \big)$$

$$v_k(y) = W_E(y)^T W_E(k)$$

Thus, the logit for token $y$ is

$$f_y^L(x) = \sum_{k=0}^{|V|-1} v_k(y) \phi_k(x)$$

Let's investigate $\phi_k(x)$. By Lemma 9,

$$\phi_k(x) = \sum_{i \in [V]} \alpha_i \left( \sum_{j \in \mathcal{N}_1(k)} W_E(j)^T W_E(i) \right)$$

$$= (b_0 - b) \sum_{j \in \mathcal{N}_1(k)} \alpha_j + \sum_{i \in [V]} \alpha_i (-a(m-2)D_H(k,i) + (b-a)m)$$

Thus, for any $k_1, k_2 \in [V]$,

$$\phi_{k_1}(x) - \phi_{k_2}(x) = (b_0 - b)\left( \sum_{j_1 \in \mathcal{N}_1(k_1)} \alpha_{j_1} - \sum_{j_2 \in \mathcal{N}_1(k_2)} \alpha_{j_2} \right)$$

$$+ \sum_{i \in [V]} \alpha_i a(m-2)(D_H(k_2,i) - D_H(k_1,i))$$

Because $-m \le D_H(k_2, i) - D_H(k_1, i) \le m$, we have

$$(b_0 - b)\left( \sum_{j_1 \in \mathcal{N}_1(k_1)} \alpha_{j_1} - \sum_{j_2 \in \mathcal{N}_1(k_2)} \alpha_{j_2} \right) - a(m-2)m$$

$$\le \phi_{k_1}(x) - \phi_{k_2}(x) \le$$

$$(b_0 - b)\left( \sum_{j_1 \in \mathcal{N}_1(k_1)} \alpha_{j_1} - \sum_{j_2 \in \mathcal{N}_1(k_2)} \alpha_{j_2} \right) + a(m-2)m$$

For prediction to be correct, we need

$$\max_{\tilde{y}} f_y^L(x) - f_{\tilde{y}}^L(x) > 0$$

This also means that

$$\max_{\tilde{y}} \sum_{k=0}^{|V|-1} \left( v_k(y) - v_k(\tilde{y}) \right) \phi_k(x) > 0$$

One can show that for any $k$, if $\iota^{-1}(\tilde{k}) = \iota^{-1}(y) \otimes \iota^{-1}(\tilde{y}) \otimes \iota^{-1}(k)$ where $\otimes$ means bitwise XOR, then

$$v_k(y) - v_k(\tilde{y}) = v_{\tilde{k}}(\tilde{y}) - v_{\tilde{k}}(y) \tag{B.1}$$

First of all, if $k = y$, then $\tilde{k} = \tilde{y}$, which means

$$v_k(y) - v_k(\tilde{y}) = v_{\tilde{k}}(\tilde{y}) - v_{\tilde{k}}(y) = b_0 + aD_H(y, \tilde{y}) - b$$

If $k \neq y, \tilde{y}$, then (B.1) implies that

$$D_H(k, y) - D_H(k, \tilde{y}) = D_H(\tilde{k}, \tilde{y}) - D_H(\tilde{k}, y)$$

We know that $D_H(k, y)$ is the number of 1s in $\iota^{-1}(k) \otimes \iota^{-1}(y)$ and,

$$\iota^{-1}(\tilde{k}) \otimes \iota^{-1}(y) = \iota^{-1}(y) \otimes \iota^{-1}(\tilde{y}) \otimes \iota^{-1}(k) \otimes \iota^{-1}(y) = \iota^{-1}(\tilde{y}) \otimes \iota^{-1}(k)$$

Similarly,

$$\iota^{-1}(\tilde{k}) \otimes \iota^{-1}(\tilde{y}) = \iota^{-1}(y) \otimes \iota^{-1}(k)$$

Therefore, (B.1) holds and we can rewrite $f_y^L(x) - f_{\tilde{y}}^L(x)$ as

$$f_y^L(x) - f_{\tilde{y}}^L(x) = \sum_{k=0}^{|V|-1} \left( v_k(y) - v_k(\tilde{y}) \right) \phi_k(x)$$

$$= (b_0 - b + aD_H(y, \tilde{y}))(\phi_y(x) - \phi_{\tilde{y}}(x))$$

$$+ \sum_{k \neq y, \tilde{y}, D_H(k,y) \ge D_H(k,\tilde{y})} a(D_H(k,y) - D_H(k,\tilde{y}))(\phi_k(x) - \phi_{\tilde{k}}(x))$$

We already know that $b_0 > b > 0$ and $a > 0$, thus, $b_0 - b + aD_H(y, \tilde{y}) > 0$ for any pair $y, \tilde{y}$.

We also want $\phi_y(x) - \phi_{\tilde{y}}(x)$ to be positive. Note that

$$\phi_y(x) - \phi_{\tilde{y}}(x) \geq (b_0 - b)(\exp(-\frac{1}{\beta}) - \exp(-\frac{2}{\beta}) - 2m\Delta) - a(m-2)m$$

We need $\Delta < \frac{\exp(-\frac{1}{\beta}) - \exp(-\frac{2}{\beta})}{2m}$ and for some positive $\Delta_1 > 0$, $b_0$ needs to be large enough such that

$$\phi_y(x) - \phi_{\tilde{y}}(x) > \Delta_1$$

which implies that

$$b_0 > \frac{a(m-2)m + \Delta_1}{\exp(-\frac{1}{\beta}) - \exp(-\frac{2}{\beta}) - 2m\Delta} + b \tag{B.2}$$

On the other hand, for $k \neq y, \tilde{y}$, we have

$$\phi_k(x) - \phi_{\tilde{k}}(x) \geq (b_0 - b)(\sum_{j_1 \in \mathcal{N}_1(k)} \alpha_{j_1} - \sum_{j_2 \in \mathcal{N}_1(\tilde{k})} \alpha_{j_2}) - a(m-2)m$$

$$\geq (b_0 - b)(-(m-1)\exp(-\frac{1}{\beta}) - \exp(-\frac{2}{\beta}) - 2m\Delta) - a(m-2)m$$

$$\geq (b_0 - b)(-(m-1)\exp(-\frac{1}{\beta}) - \exp(-\frac{2}{\beta}) + \exp(-\frac{2}{\beta}) - \exp(-\frac{1}{\beta})) - a(m-2)m$$

$$\geq -(b_0 - b)m\exp(-\frac{1}{\beta}) - a(m-2)m$$

Then, we have

$$f_y^L(x) - f_{\tilde{y}}^L(x) \geq (b_0 - b + a)\Delta_1 - \frac{|V| - 2}{2}\left((b_0 - b)am^2 \exp(-\frac{1}{\beta}) + a^2(m-2)m^2\right)$$

$$\geq \left(1 - \frac{|V| - 2}{2}am^2 \exp(-\frac{1}{\beta})\right)b_0 - (b-a)\Delta_1 + \frac{|V| - 2}{2}abm^2 \exp(-\frac{1}{\beta}) - \frac{|V| - 2}{2}a^2(m-2)m^2$$

The lower bound is independent of $\tilde{y}$, therefore, we need it to be positive to ensure the prediction is correct. To achieve this, we want

$$1 - \frac{|V| - 2}{2}am^2 \exp(-\frac{1}{\beta}) > 0$$

which implies that

$$a < \frac{2\exp(\frac{1}{\beta})}{(|V| - 2)m^2} \tag{B.3}$$

And finally we need

$$b_0 > \frac{(b-a)\Delta_1 - \frac{|V|-2}{2}abm^2 \exp(-\frac{1}{\beta}) + \frac{|V|-2}{2}a^2(m-2)m^2}{1 - \frac{|V|-2}{2}am^2 \exp(-\frac{1}{\beta})} \tag{B.4}$$

To summarize, if $b > 0$, $\Delta_1 > 0$, $0 < \Delta < \frac{\exp(-\frac{1}{\beta}) - \exp(-\frac{2}{\beta})}{2m}$, $L \geq \max\{\frac{25\log(3/\varepsilon)}{\Delta^2}, \frac{20|\mathcal{N}(y)|}{\Delta^2}\}$ for any $y$, and

$$0 < a < \frac{2\exp(\frac{1}{\beta})}{(|V| - 2)m^2}$$

and

$$b_0 > \max\{\frac{a(m-2)m + \Delta_1}{\exp(-\frac{1}{\beta}) - \exp(-\frac{2}{\beta}) - 2m\Delta} + b, \frac{(b-a)\Delta_1 - \frac{|V|-2}{2}abm^2 \exp(-\frac{1}{\beta}) + \frac{|V|-2}{2}a^2(m-2)m^2}{1 - \frac{|V|-2}{2}am^2 \exp(-\frac{1}{\beta})}\}$$

we have

$$R_{\mathcal{D}^L}(f^L) \leq \varepsilon$$

where $0 < \varepsilon < 1$.

$\square$

**Lemma 5.** *If embeddings follow (5.2) and $b = b_0$ and $\mathcal{N}(t) = V \setminus \{t\}$, then rank$(W_E) \leq m + 2$.*

*Proof.* By (5.2), we have that

$$\langle W_E(i), W_E(j) \rangle = -aD_H(i,j) + b$$

Therefore,

$$(W_E)^T W_E = -aD_H + b\mathbf{1}\mathbf{1}^T$$

Let's first look at $D_H$ which has rank at most $m + 1$. To see this, let's consider a set of $m + 1$ tokens: $\{e_0, e_1, ..., e_m\} \subseteq V$ where $e_k = 2^k$. Here $e_0$ is associated with the latent vector of all zeroes and the latent vector associated with $e_k$ has only the $k$-th latent variable being 1.

On the other hand, for any token $i$, we have that,

$$i = \sum_{k:\iota^{-1}(i)_k=1} e_k$$

In fact,

$$D_H(i) = \sum_{k:\iota^{-1}(i)_k=1} \left( D_H(e_k) - D_H(e_0) \right) + D_H(e_0)$$

where $D_H(i)$ is the $i$-th row of $D_H$, and for each entry $j$ of $D_H(i)$, we have that

$$D_H(i,j) = \sum_{k:\iota^{-1}(i)_k=1} \left( D_H(e_k,j) - D_H(e_0,j) \right) + D_H(e_0,j)$$

This is because

$$D_H(e_k,j) - D_H(e_0,j) = \begin{cases} +1 & \text{if } \iota^{-1}(j)_k = 0 \\ -1 & \text{if } \iota^{-1}(j)_k = 1 \end{cases}$$

Thus, we can rewrite $D_H(i,j)$ as

$$D_H(i,j) = \sum_{k:\iota^{-1}(i)_k=1} \left( \mathbf{1}[\iota^{-1}(i)_k = 1, \iota^{-1}(j)_k = 0] - \mathbf{1}[\iota^{-1}(i)_k = 1, \iota^{-1}(j)_k = 1)] \right) + D_H(e_0,j)$$

$$= \sum_{k=1}^{m} \left( \mathbf{1}[\iota^{-1}(i)_k = 1, \iota^{-1}(j)_k = 0] - \mathbf{1}[\iota^{-1}(i)_k = 1, \iota^{-1}(j)_k = 1)] \right)$$

$$+ \sum_{k=1}^{m} \left( \mathbf{1}[\iota^{-1}(i)_k = 0, \iota^{-1}(j)_k = 1] + \mathbf{1}[\iota^{-1}(i)_k = 1, \iota^{-1}(j)_k = 1)] \right)$$

$$= \sum_{k=1}^{m} \mathbf{1}[\iota^{-1}(i)_k = 1, \iota^{-1}(j)_k = 0] + \mathbf{1}[\iota^{-1}(i)_k = 0, \iota^{-1}(j)_k = 1]$$

$$= D_H(i,j)$$

Therefore, every row of $D_H$ can be written as a linear combination of $\{D_H(e_0), D_H(e_1), ..., D_H(e_m)\}$. In other words, $D_H$ has rank at most $m + 1$.

Therefore,

$$\text{rank}((W_E)^T W_E) = \text{rank}(W_E) \leq m + 2.$$

$\square$

**Lemma 9.** *Let $z^{(0)}$ and $z^{(1)}$ be two binary vectors of size $m$ where $m \geq 2$. Then,*

$$\sum_{z:D_H(z^{(0)},z)=1} D_H(z, z^{(1)}) = (m-2)D_H(z^{(0)}, z^{(1)}) + m$$

*Proof.* For $z$ such that $D_H(z, z^{(0)}) = 1$, we know that there are two cases. Either $z$ differs with $z^{(0)}$ on a entry but agrees with $z^{(1)}$ on that entry or $z$ differs with both $z^{(0)}$ and $z^{(1)}$.

For the first case, we know that there are $D_H(z^{(0)}, z^{(1)})$ such entries. In this case, $D_H(z, z^{(1)}) = D_H(z^{(0)}, z^{(1)}) - 1$. For the second case, $D_H(z, z^{(1)}) = D_H(z^{(0)}, z^{(1)}) + 1$.

Therefore,

$$\sum_{z:D_H(z,z^{(0)})=1} D_H(z, z^{(1)})$$
$$= D_H(z^{(0)}, z^{(1)})(D_H(z^{(0)}, z^{(1)}) - 1) + (m - D_H(z^{(0)}, z^{(1)}))(D_H(z^{(0)}, z^{(1)}) + 1)$$
$$= (m - 2)D_H(z^{(0)}, z^{(1)}) + m$$

□

**Lemma 10.** *If $m \geq 3$ and $\mathcal{N}(t) = V \setminus \{t\}$, then $\mathcal{N}_1(t) \not\subseteq \mathcal{N}_1(t')$ for any $t, t' \in [V]$.*

*Proof.* For any token $t$, $\mathcal{N}_1(t)$ contains any token $t'$ such that $D_H(t, t') = 1$ by the conditions. Then given a set $\mathcal{N}_1(t)$, one can uniquely determine token $t$. This is because for the set of latent vectors associated with $\mathcal{N}_1(t)$, at each index, there could only be one possible change. □

## B.4 Proofs for Section 5.4

**Lemma 6.** *Suppose the data generating process follows Section 4.1 and $\mathcal{N}(z^*) = \{z : z_1^* = z_1\} \setminus \{z^*\}$. Given the last token in the sequence $t_L$, then*

$$\nabla_{u_{t,t_L}} \ell(f^L) = \nabla \ell(f^L)^T (W_E)^T W^V (\alpha_t \hat{p}_t W_E(t) - \hat{p}_t \sum_{l=1}^{L} \hat{p}_{t_l} W_E(t_l))$$

*where for token $t$, $\alpha_t = \sum_{l=1}^{L} \mathbf{1}[t_l = t]$ and $\hat{p}_t$ is the normalized attention score for token $t$.*

*Proof.* Recall that,

$$f^L(x) = \left[ W_E^T W_V \, \text{attn}(W_E \chi(x)) \right]_{:L}$$
$$= W_E^T W_V \sum_{l=1}^{L} \frac{\exp(u_{t_l, t_L})}{Z} W_E(t_l)$$

where $Z$ is a normalizing constant.

Define $\hat{p}_{t_l} = \frac{\exp(u_{t_l, t_L})}{Z}$. Then we have

$$f^L(x) = W_E^T W_V \sum_{l=1}^{L} \hat{p}_{t_l} W_E(t_l)$$

Note that if $t_l = t$ then,

$$\frac{\partial \hat{p}_{t_l}}{\partial u_{t,t_L}} = \hat{p}_{t_l}(1 - \hat{p}_{t_l})$$

Otherwise,

$$\frac{\partial \hat{p}_{t_l}}{\partial u_{t,t_L}} = -\hat{p}_{t_l} \hat{p}_t$$

By the chain rule, we know that

$$\nabla_{u_{t,t_L}} \ell(f^L) = \nabla \ell(f^L)^T (W_E)^T W^V (\sum_{l=1}^{L} \mathbf{1}[t_l = t] \hat{p}_{t_l} W_E(t) - \sum_{l=1}^{L} \hat{p}_{t_l} \hat{p}_t W_E(t_l))$$

Therefore,

$$\nabla_{u_{t,t_L}} \ell(f^L) = \nabla \ell(f^L)^T (W_E)^T W^V (\alpha_t \hat{p}_t W_E(t) - \hat{p}_t \sum_{l=1}^{L} \hat{p}_{t_l} W_E(t_l))$$

where $\alpha_t = \sum_{l=1}^{L} \mathbf{1}[t_l = t]$. □

# C   Additional experiments – context hijacking

In this section, we show the results of additional context hijacking experiments on the COUNTERFACT dataset [Men+22].

**Reverse context hijacking**   In Figure 2a, we saw the effects of hijacking by adding in "Do not think of {target_false}." to each context. Now, we measure the effect of the reverse: What if we prepend "Do not think of {target_true}." ?

Based on the study in this paper on how associative memory works in LLMs, we should expect the efficacy score to decrease. Indeed, this is what happens, as we see in Figure C.1.

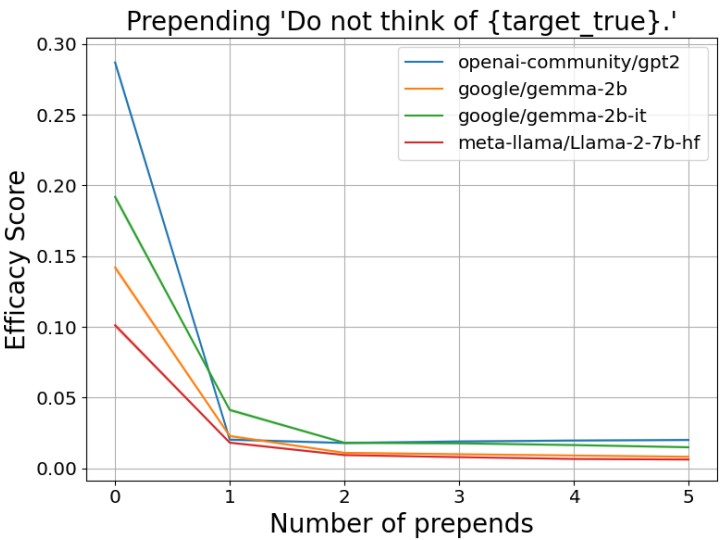

**Figure C.1:** Prepending 'Do not think of {target_true}.' can increase the chance of LLMs to output correct tokens. This figure shows efficacy score versus the number of prepends for various LLMs on the COUNTERFACT dataset with the reverse context hijacking scheme.

**Hijacking based on relation IDs**   We first give an example of each of the 4 relation IDs we hijack in Table 1.

**Table 1:** Examples of contexts in Relation IDs from COUNTERFACT

| RELATION ID $r$ | CONTEXT $p$ | TRUE TARGET $o_*$ | FALSE TARGET $o_-$ |
|---|---|---|---|
| P190 | Kharkiv is a twin city of | Warsaw | Athens |
| P103 | The native language of Anatole France is | French | English |
| P641 | Hank Aaron professionally plays the sport | baseball | basketball |
| P131 | Kalamazoo County can be found in | Michigan | Indiana |

**Table 2:** Examples of hijack and reverse hijack formats based on Relation IDs

| RELATION ID $r$ | CONTEXT HIJACK SENTENCE | REVERSE CONTEXT HIJACK SENTENCE |
|---|---|---|
| P190 | The twin city of {subject} is not {target_false} | The twin city of {subject} is {target_true} |
| P103 | {subject} cannot speak {target_false} | {subject} can speak {target_true} |
| P641 | {subject} does not play {target_false} | {subject} plays {target_true} |
| P131 | {subject} is not located in {target_false} | {subject} is located in {target_true} |

Similar to Figure 2b, we repeat the hijacking experiments where we prepend factual sentences generated from the relation ID. We use the format illustrated in Table 2 for the prepended sentences.

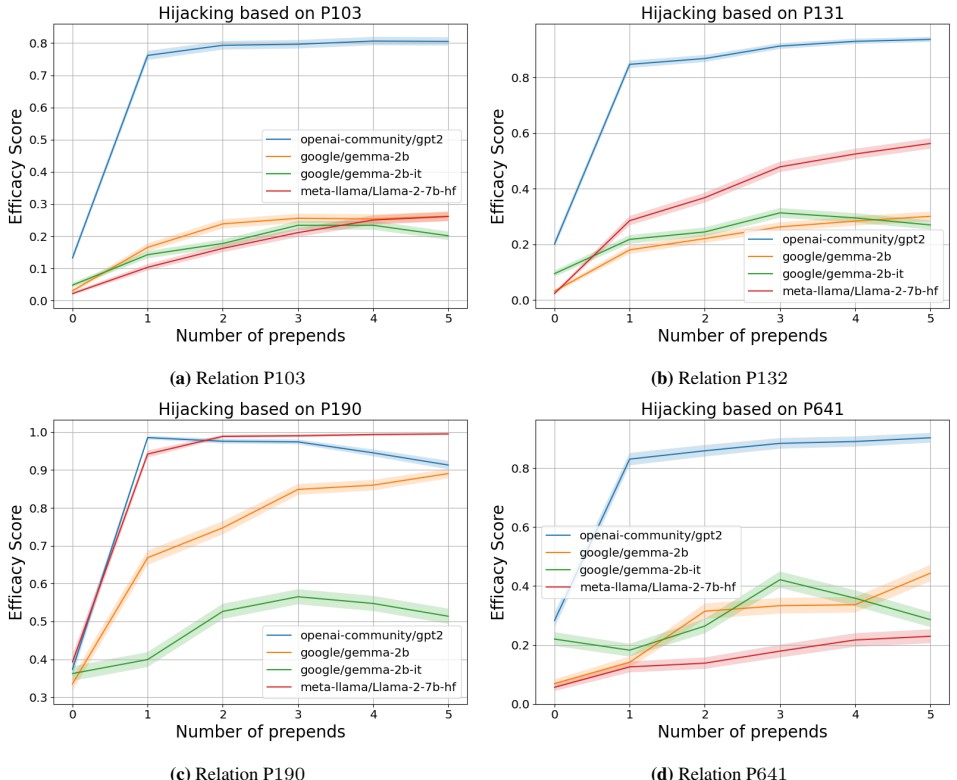

**Figure C.2:** Context hijacking based on relation IDs can result in LLMs output incorrect tokens. This figure shows efficacy score versus the number of prepends for various LLMs on the COUNTERFACT dataset with hijacking scheme presented in Table 2.

We experiment with 3 other relation IDs and we see similar trends for all the LLMs in Figure C.2a, C.2b, and C.2d. That is, the efficacy score rises for the first prepend and as we increase the number of prepends, the trend of ES rising continues. Therefore, this confirms our intuition that LLMs can be hijacked by contexts without changing the factual meaning.

Similar to Figure C.1, we experiment with reverse context hijacking where we give the answers based on relation IDs, as shown in Table 2. We again experiment with the same 4 relation IDs and the results are in Figure C.3a - C.3d. We see that the efficacy score decreases when we prepend the answer sentence, thereby verifying the observations of this study.

**Hijacking without exact target words** So far, the experiments use prompts that either contain true or false target words. It turns out, the inclusion of exact target words are not necessary. To see this, we experiment a variant of the generic hijacking and reverse hijacking experiments. But instead of saying "Do not think of {target_false}" or "Do not think of {target_true}". We replace target words with words that are semantically close. Specifically, for relation P1412, we replace words representing language (e.g., "French") with their associated country name (e.g., "France"). As shown in Figure C.4, context hijacking and reverse hijacing still work in this case.

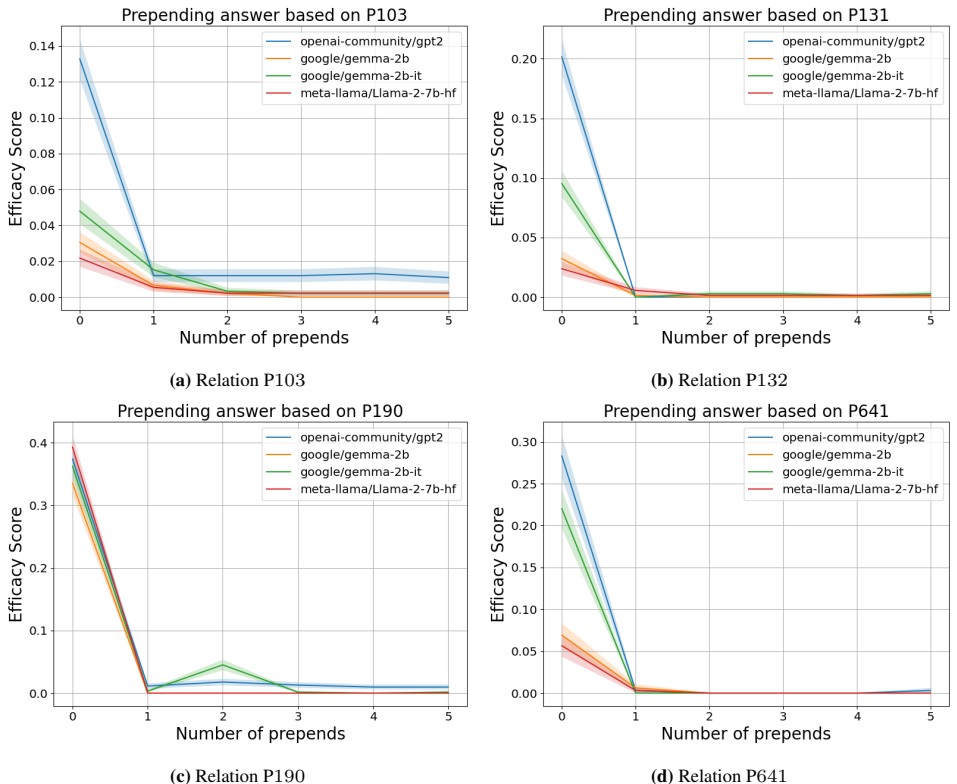

**(a)** Relation P103

**(b)** Relation P132

**(c)** Relation P190

**(d)** Relation P641

**Figure C.3:** Reverse context hijacking based on relation IDs can result in LLMs to be more likely to be correct. This figure shows efficacy score versus the number of prepends for various LLMs on the COUNTERFACT dataset with the reverse hijacking scheme presented in Table 2.

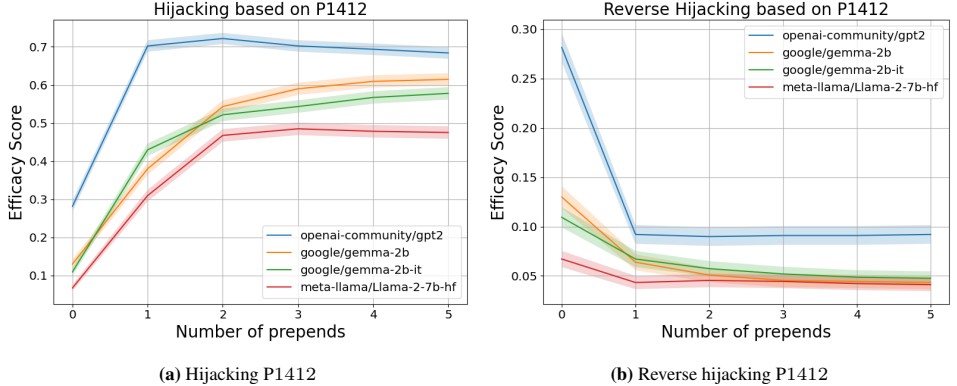

**(a)** Hijacking P1412

**(b)** Reverse hijacking P1412

**Figure C.4:** Hijacking and reverse hijacking experiments on relation P1412 show that context hijacking does not require exact target word to appear in the context. This figure shows efficacy score versus the number of prepends for various LLMs on the COUNTERFACT dataset.

# D  Additional experiments and figures – latent concept association

In this appendix section, we present additional experimental details and results from the synthetic experiments on latent concept association.

**Experimental setup**  Synthetic data are generated following the model in Section 4.1. Unless otherwise stated, the default setup has $\omega = 0.5$, $\beta = 1$ and $\mathcal{N}(i) = V \setminus \{i\}$ and $L = 256$. The default hidden dimension of the one-layer transformer is also set to be 256. The model is optimized using AdamW [LH17] where the learning rate is chosen from $\{0.01, 0.001\}$. The evaluation dataset

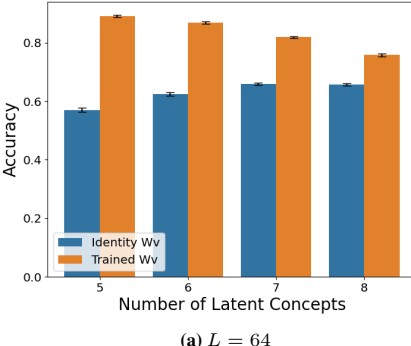
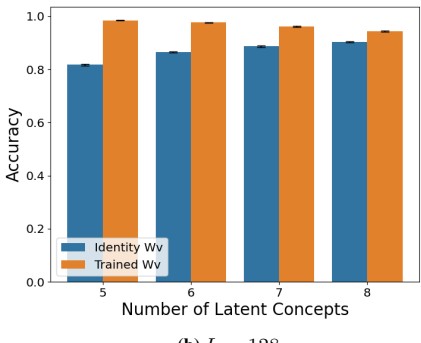

**(a)** $L = 64$            **(b)** $L = 128$

**Figure D.1:** Fixing the value matrix $W_V$ as the identity matrix results in lower accuracy compared to training $W_V$, especially for smaller context length $L$. The figure reports accuracy for both fixed and trained $W_V$ settings, with standard errors calculated over 10 runs.

is drawn from the same distribution as the training dataset and consists of 1024 $(x, y)$ pairs. Although theoretical results in Section 5 may freeze certain parts of the network for simplicity, in this section, unless otherwise specified, all layers of the transformers are trained jointly. Also, in this section, we typically report accuracy which is $1 -$ error.

## D.1    On the value matrix $W_V$

In this section, we provide additional figures of Section 6.1. Specifically, Figure D.1 shows that fixing the value matrix to be the identity will negatively impact accuracy. Figure D.2 indicates that replacing trained value matrices with constructed ones can preserve accuracy to some extent. Figure D.3 suggests that trained value matrices and constructed ones share similar low-rank approximations. For the last two sets of experiments, we consider randomly constructed value matrix, where the outer product pairs are chosen randomly, defined formally as follows:

$$W_V = \sum_{i \in [V]} W_E(i) \left( \sum_{\{j\} \sim \text{Unif}([V])^{|\mathcal{N}_1(i)|}} W_E(j)^T \right)$$

## D.2    On the embeddings

This section provides additional figures from Section 6.2. Figure D.4 shows that in the underparameterized regime, embedding training is required. Figure D.5 indicates that the embedding structure in the underparameterized regime roughly follows (5.2). Finally Figure D.6 shows that, when the value matrix is fixed to the identity, the relationship between inner product of embeddings and their corresponding Hamming distance is mostly linear.

## D.3    On the attention selection mechanism

This section provides additional figures from Section 6.3. Figure D.7-D.8 show that attention mechanism selects tokens in the same cluster as the last token. In particular, for Figure D.8, we extend experiments to consider cluster structures that depend on the first two latent variables. In other words, for any latent vector $z^*$, we have

$$\mathcal{N}(z^*) = \{z : z_1^* = z_1 \text{ and } z_2^* = z_2\} \setminus \{z^*\}$$

## D.4    Spectrum of embeddings

We display several plots of embedding spectra (Figure D.9, Figure D.10, Figure D.11, Figure D.12) that exhibit eigengaps between the top and bottom eigenvalues, suggesting low-rank structures.

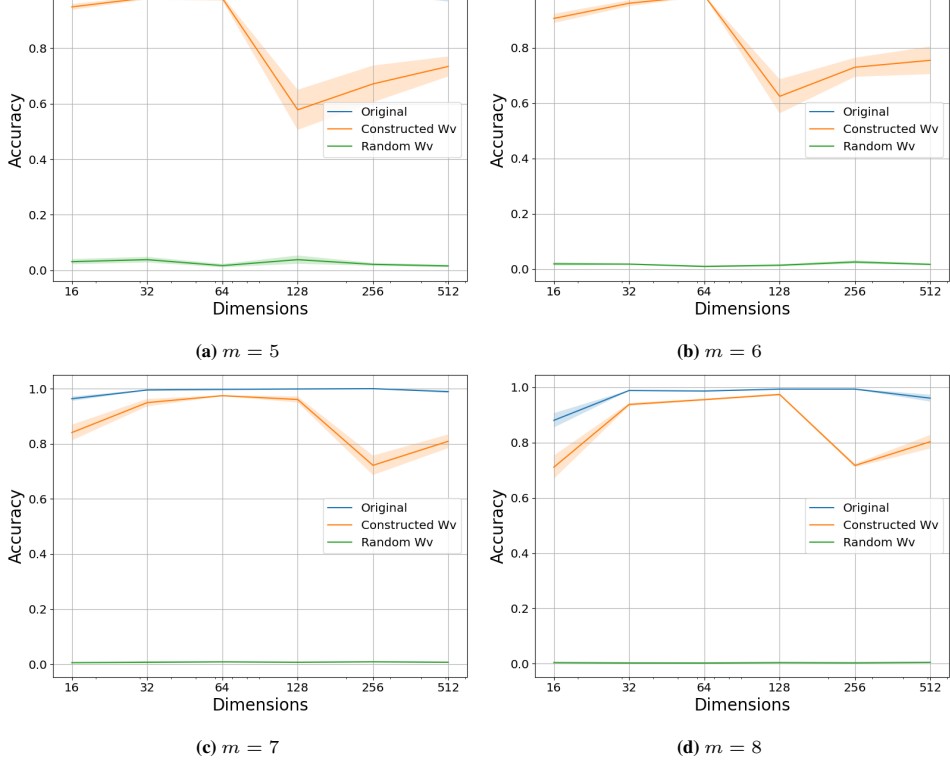

**Figure D.2:** When the value matrix is replaced with the constructed one in trained transformers, the accuracy does not significantly decrease compared to replacing the value matrix with randomly constructed ones. The graph reports accuracy under different embedding dimensions and standard errors are over 5 runs.

### D.5 Context hijacking in latent concept association

In this section, we want to simulate context hijacking in the latent concept association model. To achieve that, we first sample two output tokens $y^1$ (true target) and $y^2$ (false target) and then generate contexts $x^1 = (t^1_1, ..., t^1_L)$ and $x^2 = (t^2_1, ..., t^2_L)$ from $p(x^1|y^1)$ and $p(x^2|y^2)$. Then we mix the two contexts with rate $p_m$. In other words, for the final mixed context $x = (t_1, ..., t_L)$, $t_l$ has probability $1 - p_m$ to be $t^1_l$ and $p_m$ probability to be $t^2_l$. Figure D.13 shows that, as the mixing rate increases from 0.0 to 1.0, the trained transformer tends to favor predicting false targets. This mirrors the phenomenon of context hijacking in LLMs.

### D.6 On the context lengths

As alluded in Section 5.5, the memory recall rate is closely related to the KL divergences between context conditional distributions. Because contexts contain mostly i.i.d samples, longer contexts imply larger divergences. This is empirically verified in Figure D.14 which demonstrates that longer context lengths can lead to higher accuracy.

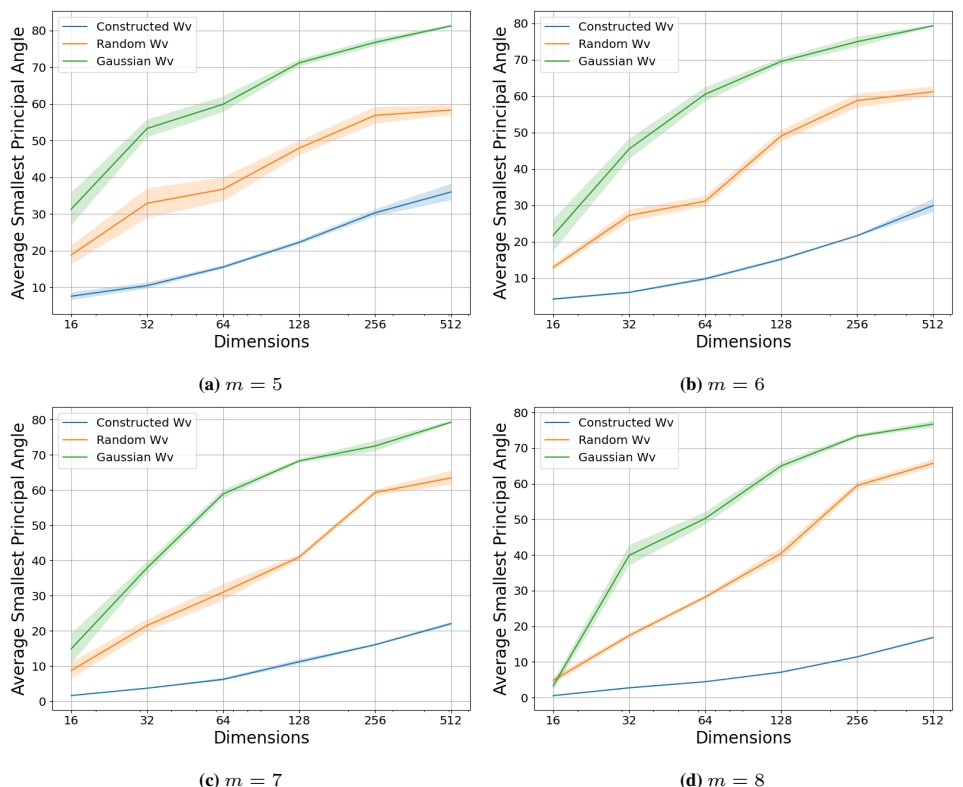

**(a)** $m = 5$

**(b)** $m = 6$

**(c)** $m = 7$

**(d)** $m = 8$

**Figure D.3:** The constructed value matrix $W_V$ has similar low rank approximation with the trained value matrix. The figure displays average smallest principal angles between low-rank approximations of trained value matrices and those of constructed, randomly constructed, and Gaussian-initialized value matrices. Standard errors are over $5$ runs.

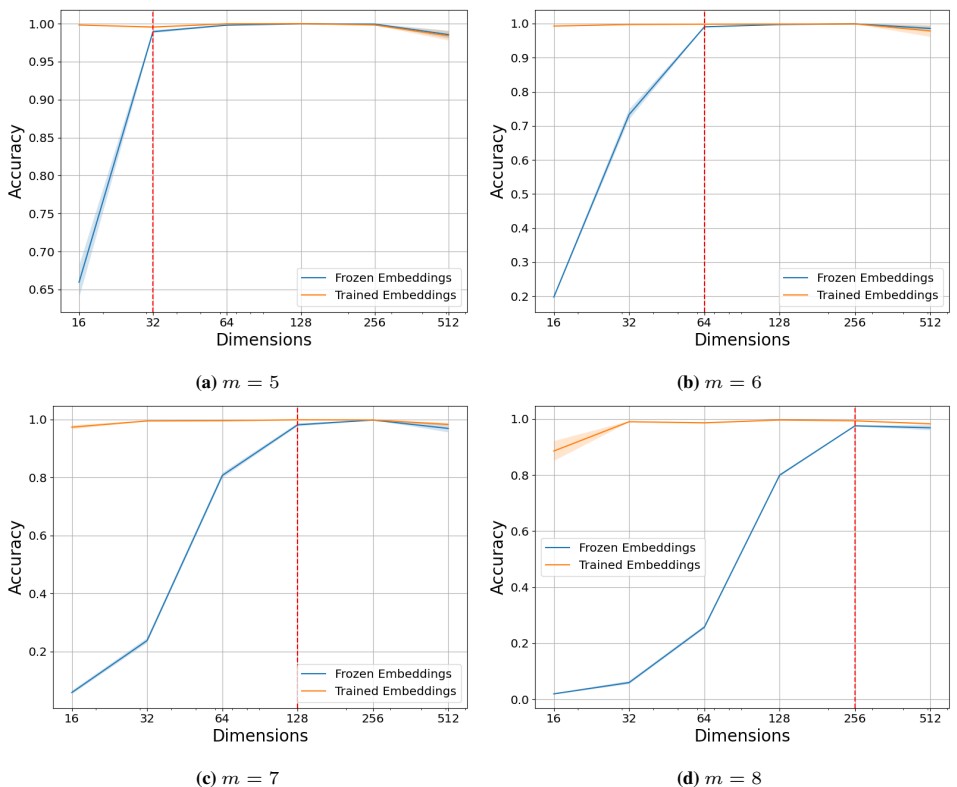

**Figure D.4:** In the underparameterized regime $(d < V)$, freezing embeddings to initializations causes a significant decrease in performance. The graph reports accuracy with different embedding dimensions and the standard errors are over 5 runs. Red lines indicate when $d = V$.

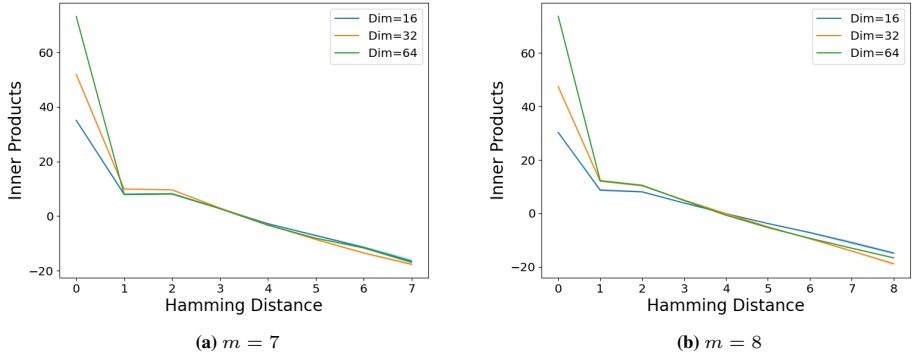

**Figure D.5:** The relationship between inner products of embeddings and corresponding Hamming distances of tokens can be approximated by (5.2). The graph displays the average inner product between embeddings of two tokens against the corresponding Hamming distance between these tokens. Standard errors are over 5 runs.

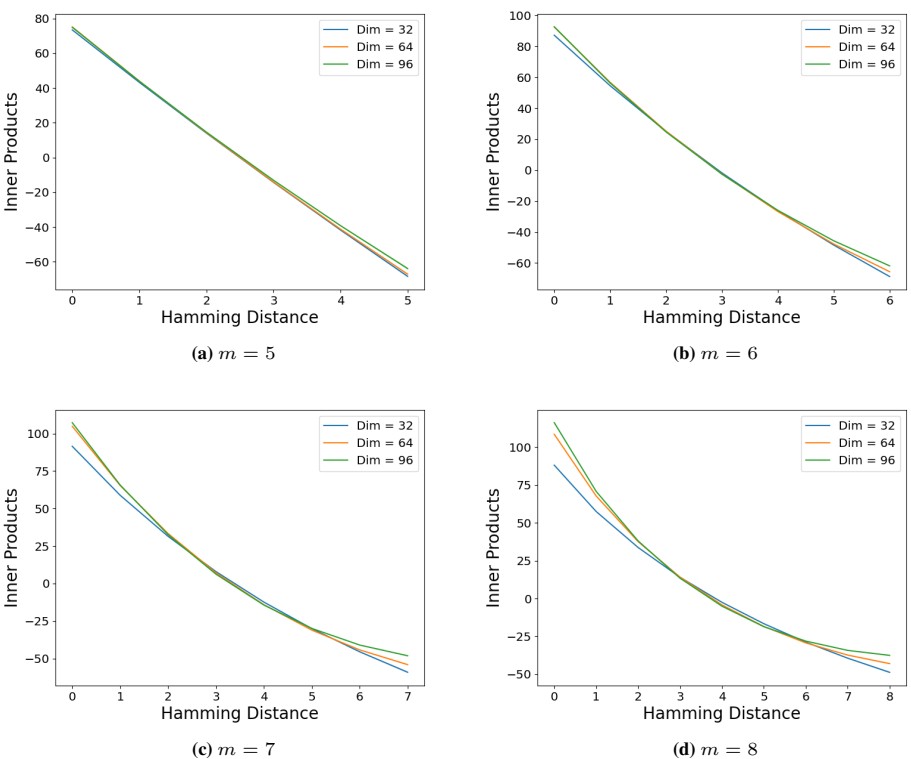

**Figure D.6:** The relationship between inner products of embeddings and corresponding Hamming distances of tokens is mostly linear when the value matrix $W_V$ is fixed to be the identity. The graph displays the average inner product between embeddings of two tokens against the corresponding Hamming distance between these tokens. Standard errors are over 10 runs.

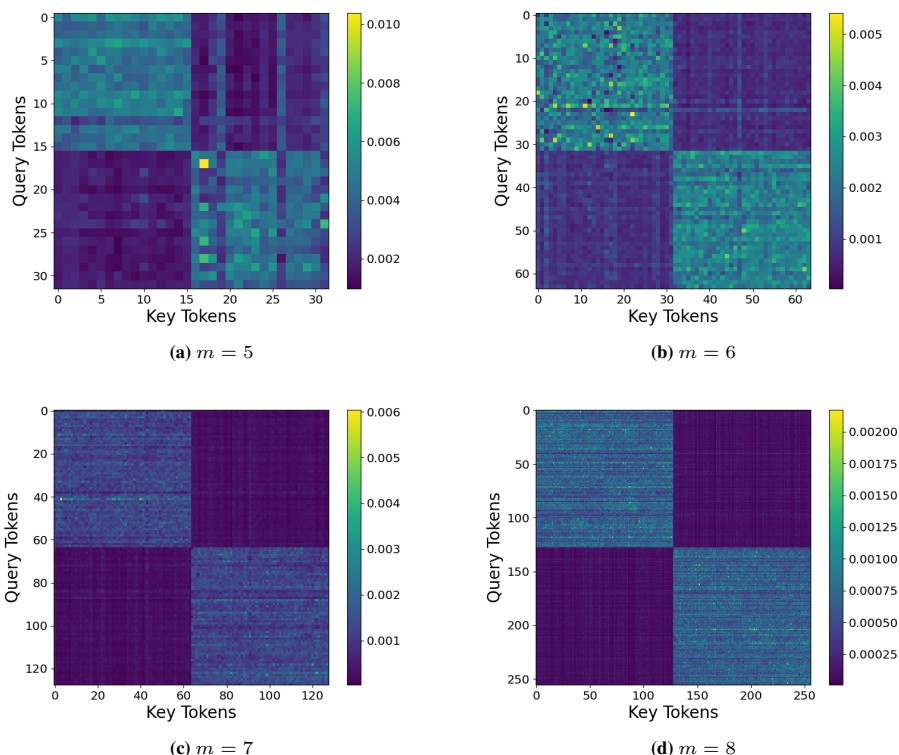

**Figure D.7:** The attention patterns show the underlying cluster structure of the data generating process. Here, for any latent vector, we have $\mathcal{N}(z^*) = \{z : z_1^* = z_1\} \setminus \{z^*\}$. The figure shows attention score heat maps that are averaged over 10 runs.

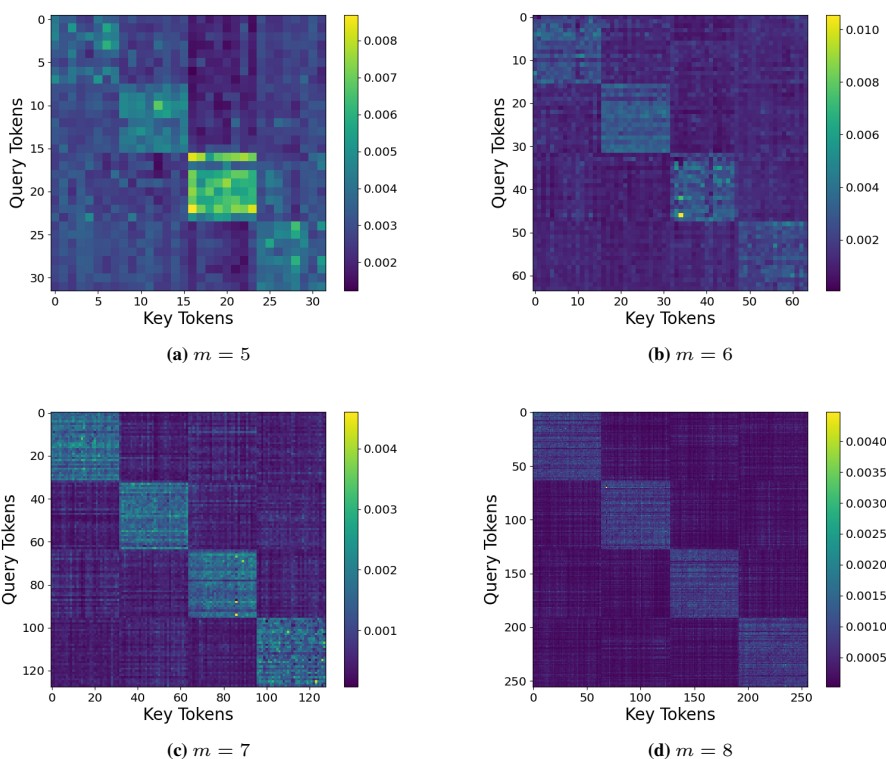

**Figure D.8:** The attention patterns show the underlying cluster structure of the data generating process. Here, for any latent vector, we have $\mathcal{N}(z^*) = \{z : z_1^* = z_1 \text{ and } z_2^* = z_2\} \setminus \{z^*\}$. The figure shows attention score heat maps that are averaged over 10 runs.

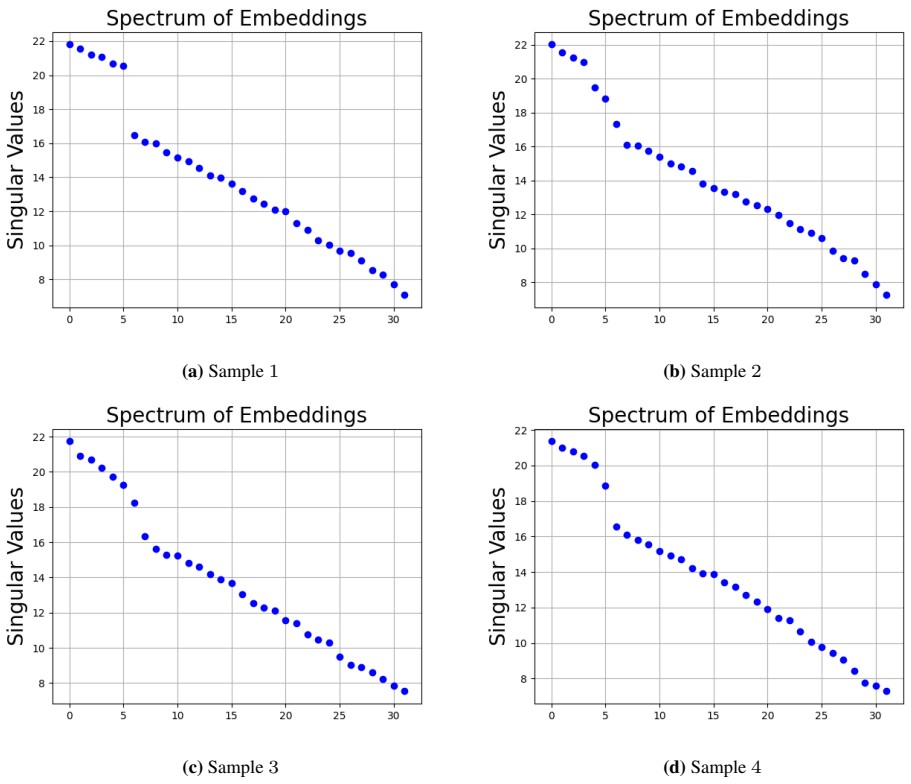

**Figure D.9:** The spectrum of embedding matrix $W_E$ has eigengaps between the top and bottom eigenvalues, indicating low rank structures. The figure shows results from 4 experimental runs. Number of latent variable $m$ is 7 and the embedding dimension is 32.

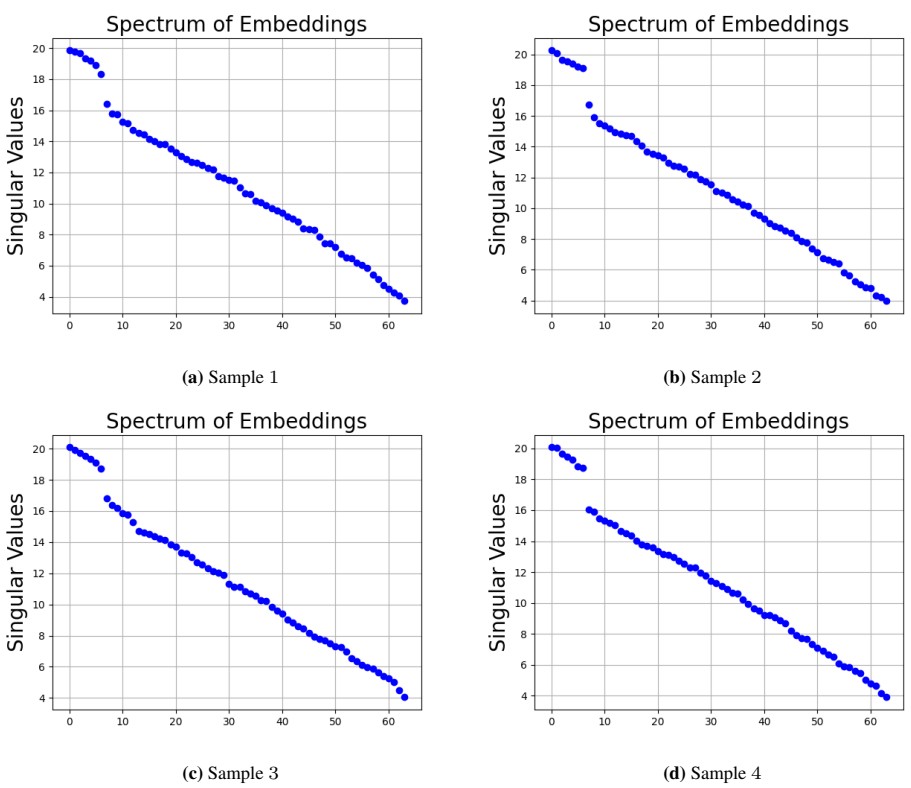

**Figure D.10:** The spectrum of embedding matrix $W_E$ has eigengaps between the top and bottom eigenvalues, indicating low rank structures. The figure shows results from 4 experimental runs. Number of latent variable $m$ is 7 and the embedding dimension is 64.

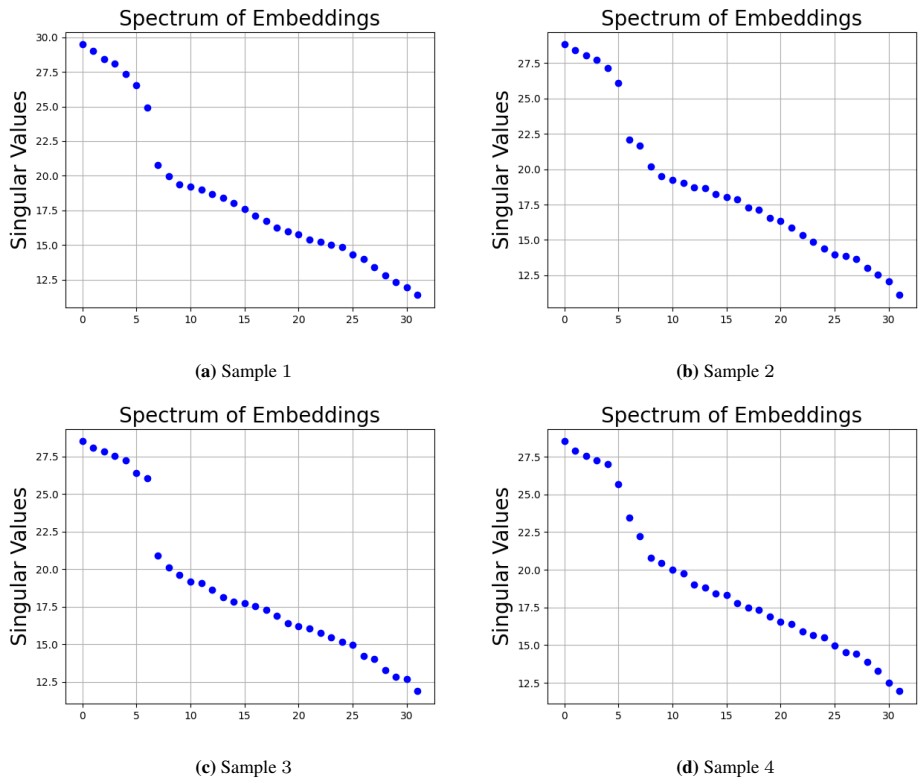

**Figure D.11:** The spectrum of embedding matrix $W_E$ has eigengaps between the top and bottom eigenvalues, indicating low rank structures. The figure shows results from 4 experimental runs. Number of latent variable $m$ is 8 and the embedding dimension is 32.

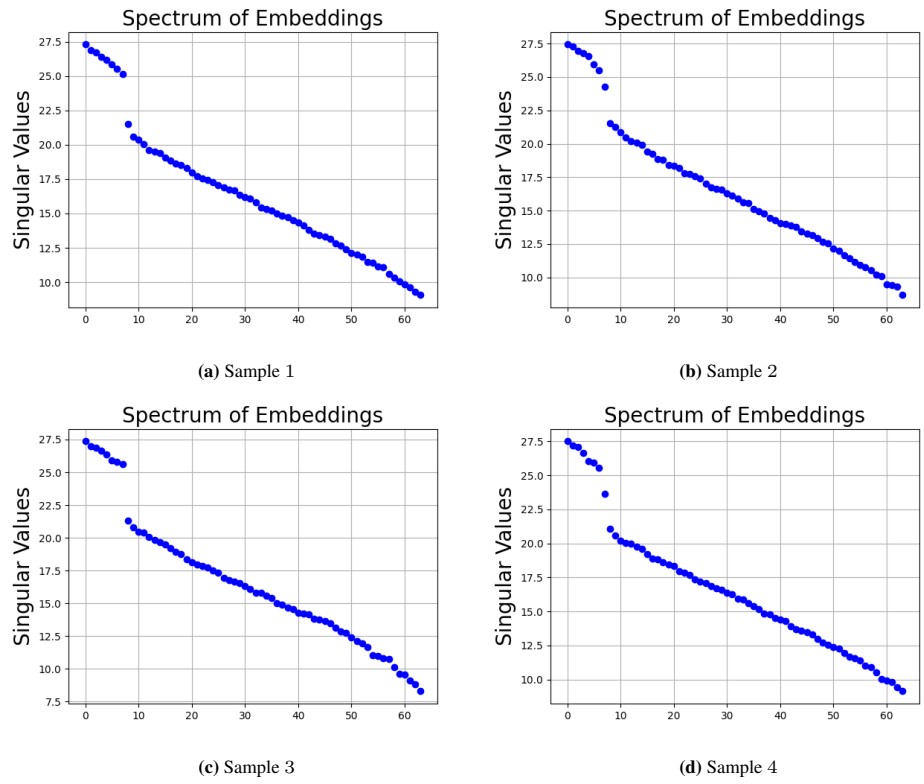

**(a)** Sample 1

**(b)** Sample 2

**(c)** Sample 3

**(d)** Sample 4

**Figure D.12:** The spectrum of embedding matrix $W_E$ has eigengaps between the top and bottom eigenvalues, indicating low rank structures. The figure shows results from 4 experimental runs. Number of latent variable $m$ is 8 and the embedding dimension is 64.

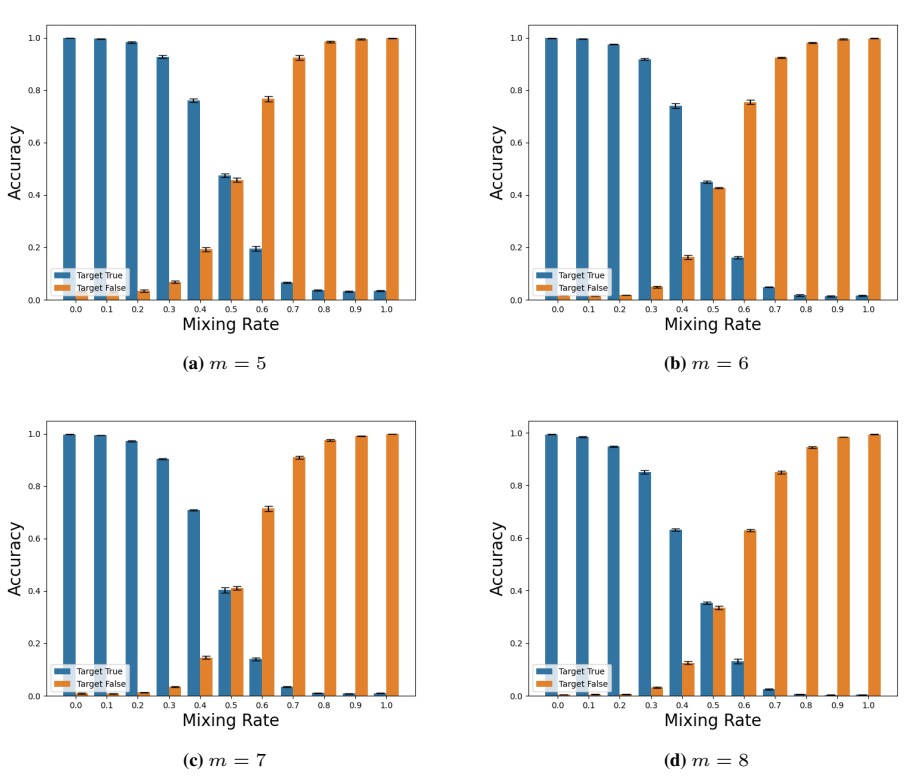

**Figure D.13:** Mixing contexts can cause misclassification. The figure reports accuracy for true target and false target under various context mixing rate. Standard errors are over 5 runs.

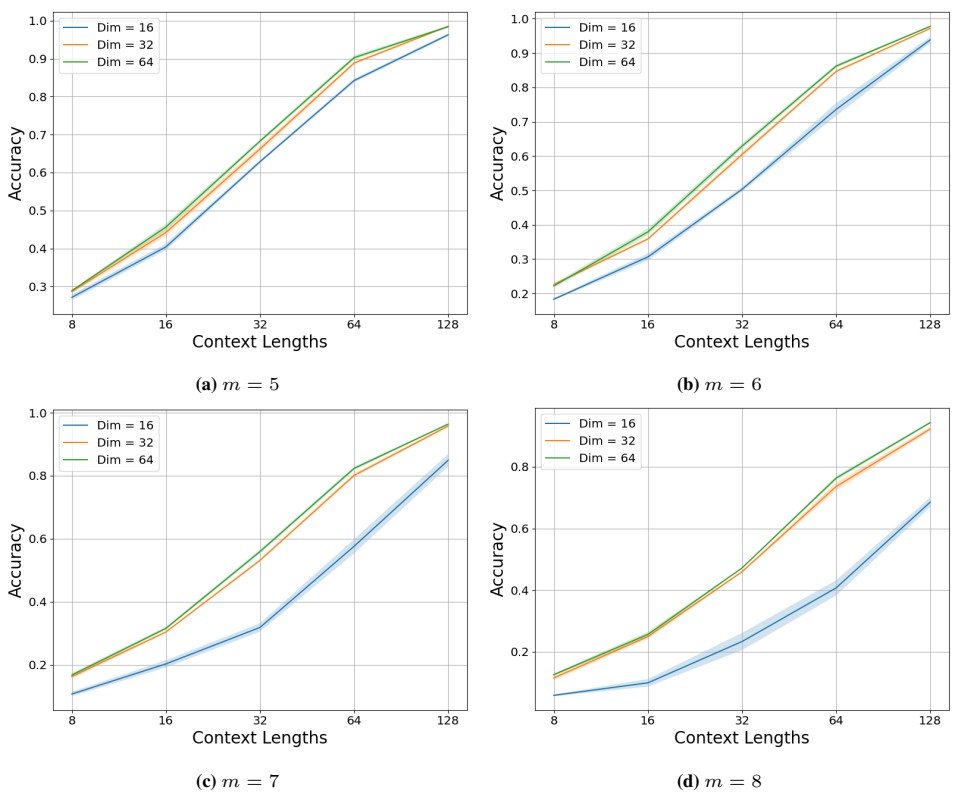

**(a)** $m = 5$

**(b)** $m = 6$

**(c)** $m = 7$

**(d)** $m = 8$

**Figure D.14:** Increasing context lengths can improve accuracy. The figure reports accuracy across various context lengths and dimensions. Standard errors are over 5 runs.

