# OpenReview forum: "Do LLMs dream of elephants (when told not to)? Latent concept association and associative memory in transformers"
_NeurIPS.cc/2024/Conference — NeurIPS 2024 poster_

### Official Review · Reviewer_MLxQ · 2024-07-03

**Soundness:** 3
**Presentation:** 3
**Contribution:** 3
**Rating:** 6
**Confidence:** 3

**Summary:**

This paper demonstrates the phenomenon of "context hijacking" in LLMs, where repeated mentions of sentences in the context could negatively influence a model's factual recall. Motivated by this, the authors then formulate an associative memory recall task and prove theoretically certain properties of one-layer transformer models on this task, which shows that transformers are capable of performing the task and the roles of their individual components in the task. Further experiments verify the results.

**Strengths:**

- Studying how language models' generation is influenced by the context is an important direction toward better understanding and improving LLMs (e.g., hallucinations). The proposed context hijacking is interesting, and could be thought of as another way of stress testing LMs.

- Analyzing the theoretical properties of transformers in associative memory recall tasks could inspire future work and formations toward a better understanding of transformers and their actuality. The roles of different components in the proposed task also help open up the black box of transformers and could potentially inspire future investigations of the role of real LMs' components.

- The paper is generally well-written.

**Weaknesses:**

- It is unclear to what degree the context hijacking phenomenon still exists in more powerful models (e.g., GPT-4) with better semantic understanding. For example, I tried the proposed attack methods on GPT-4o and it never influenced the model negatively.

- Related to the last point, the proposed associative memory formation lacks modeling of semantics, where the task boils down to tokens and their similarity. This may be a good formulation for models that are not powerful enough and mostly rely on surface cues to generate the next token, but not for models that have strong language understanding and logic. This also relates to the shallow network (one-layer transformer) which is the main target for theoretical analysis. The results may not transfer/be enlightening for improving the current LLMs.

**Questions:**

None

**Limitations:**

It would be better to include a dedicated "limitations" section beyond those in the checklist.

---

> ### Author Rebuttal · Authors · 2024-08-07
>
> Thanks for your support and useful comments! Your suggestion is really valuable in helping us clarify our paper.
>
>
> **_Questions on larger models like GPT-4 and limitations of single-layer transformer_**
>
>
> This is a great question! First of all, we didn’t test this on GPT-4 because, as a closed-source model, it’s unclear what is behind its predictions. For example, what kind of instruction tuning is trained and does it rely on external sources (web search + rag) for basic fact prediction.
>
>
> Nonetheless, even in the official GPT-4 technical report [1], we see an example similar to context hijacking (Elvis Perkins example). In that example, the prompt is “Son of an actor, this American guitarist and rock singer released many songs and albums and toured with his band. His name is "Elvis" what?”. GPT-4 answers with Presley, even though the answer is Perkins (Elvis Presley is not the son of an actor). GPT-4 can be viewed as distracted by all the information related to music and answers Presley. In fact, it is known that LLMs can be easily distracted by contexts in use cases other than fact retrieval such as problem-solving [2] which we reference many in the paper (L78-82, L120-121). So we reasonably suspect that similar behavior still exists in larger models but is harder to exploit. This is similar to the “Chicago'' example we give in Figure 1, where a larger model like LLaMA-7B requires more prepends but is still hackable and we provide more detailed experiments in section 3 as well.
>
>
> On the other hand, in the literature, theoretical works on multi-layer transformers are limited and remain an active research area while studying single-layer transformers allows us to easily do carefully controlled experiments. Even with single-layer transformers, we have obtained many interesting results, such as approximated value matrix structure (L331-L343) and low-rank structure of embeddings (L359-L362). In particular, the approximated value matrix structure is closely related to earlier works on multi-layer transformers as well [3]. Moreover, the existence of the low-rank structure provides the theoretical grounding for many existing editing and fine-tuning methods including LORA and ROME that exploit low-rank structure. A lot of these methods work on current LLMs. Note that such low-rank structures naturally emerge by just training single-layer transformers. We hope these results can further lay the foundation for future research on multi-layer transformers on similar topics.
>
>
>
>
> [1] Achiam, Josh, et al. "Gpt-4 technical report." arXiv preprint arXiv:2303.08774 (2023).
>
>
> [2] Shi, Freda, et al. "Large language models can be easily distracted by irrelevant context." International Conference on Machine Learning. PMLR, 2023.
>
>
> [3] Bietti, Alberto, et al. "Birth of a transformer: A memory viewpoint." Advances in Neural Information Processing Systems 36 (2024).

---

> ### Comment · Reviewer_MLxQ · 2024-08-10
>
> Thank you for the response, which addresses my concerns to some degree. I will raise my evaluation.

---

> > ### Author Response · Authors · 2024-08-11
> >
> > We are pleased that your concerns have been addressed. We truly appreciate your time and effort to engage with our work, and for updating your score accordingly.

---

### Official Review · Reviewer_vgcR · 2024-07-09

**Soundness:** 3
**Presentation:** 3
**Contribution:** 4
**Rating:** 7
**Confidence:** 3

**Summary:**

This paper studies the mechanics of factual recall in transformer-based models, framing the problem as next token prediction. In particular, the authors focus on the brittleness of language models, which can be elicited to provide different answers to factual queries by adding distracting information in the prompt (a procedure which the authors term context hijacking). This phenomenon is shown for various language models, with sizes up to 7B parameters.
The authors then formulate a hypothesis for the hijacking phenomenon, according to which the model predicts the next token based on a similarity measure in a latent concept space. They provide a series of theoretical results that explain how a single-layer transformer can solve the latent concept association problem.
Finally, the paper includes empirical validation of the theoretical results.

**Strengths:**

- The problem studied is important to improve our understanding of language models' internal mechanisms.
- The paper presents an interesting theoretical analysis of factual recall and provides solid empirical evidence to support it.

**Weaknesses:**

- The authors do not discuss any limitations of their work. For instance, the authors assume that each latent concept is associated to one token only, how would the theoretical results look without this assumption? Moreover, the study is motivated by the behavior of multi-billion-parameter models but focuses on a single-layer transformer, how do the authors expect their results to generalize to larger models?
- In the motivation of the problem, the authors show how LLaMA 7B can be “hijacked” to output a wrong answer to the prompt about the location of the Eiffel Tower. However, a text consisting of the sentence “The Eiffel Tower is not in Chicago” repeated eight times represents an input arguably out of distribution with respect to the model’s pre-training data. In this setting, it is possible that the model continues the prompt ”the Eiffel Tower is in the city of” using a different mechanism than it would have without the prepended hijacking context. A test for the authors’ hypothesis would be to prepend a non-templated paragraph about, for example, the city of Chicago (possibly mentioning that the Eiffel Tower is not located there). If the authors’ hypothesis is correct, this should still steer the model towards completing “the Eiffel Tower is in the city of” with “Chicago.” Is this the case?
- Minor point: the figures can be improved (e.g., increasing the font size)

**Questions:**

- Maybe I am missing something, but isn’t the efficacy score expected to increase as the hijacking context gets longer (i.e., the more times the sentence “Do not think of {target_false}.” gets repeated, the more likely the model is to assign higher probability mass to target_false)?

**Limitations:**

The authors do not address any limitation of their work. Their answer to the checklist question "Does the paper discuss the limitations of the work performed by the authors?" the authors motivate their "yes" answer with a single sentence: "Studying single-layer transformers is limited."

---

> ### Author Rebuttal · Authors · 2024-08-07
>
> Thank you for your positive feedback and questions. We're glad that you found the paper to be important, interesting, and solid!
>
>
> **_Limitations_**
>
> Sorry for the confusion! We don't assume that each latent concept is associated with only one token. Instead, the latent concept space consists of multiple latent variables, with each latent vector associated with one token. Therefore, each token can represent multiple latent concepts.
>
> The study is motivated by the behavior of multi-billion-parameter models. But this behavior – context hijacking – is a failure case. It is thus reasonable to expect this behavior to translate to smaller models as well. However, studying single-layer transformers allows us to conduct detailed, controlled experiments. Moreover, current theoretical work on multi-layer transformers is limited and remains an active research area. Even with single-layer transformers, we have obtained many interesting results including approximated value matrix structure (L331-L343) and low-rank structure of embeddings (L359-L362). In particular, the existence of the low-rank structure provides the theoretical grounding for many editing and fine-tuning methods including LORA and ROME that exploit low-rank structure. Note that such low-rank structures naturally emerge by just training single-layer transformers.
>
> **_Question on templated hijacking prompt_**
>
> This is a great question! First of all, in our systematic experiments, we saw that even after one prepend, arguably more in line with the training distributions and less templated, there's still a performance downgrade.
>
> But **to answer your question,** we conducted a new test by placing a description of Chicago from Wikipedia at the beginning of the prompt: "Chicago is the most populous city in the U.S. state of Illinois and in the Midwestern United States. With a population of 2,746,388 as of the 2020 census, it is the third-most populous city in the United States. Therefore, The Eiffel Tower is in the city of". When presented with this prompt, all four models responded with "Chicago." This suggests the hypothesis still makes sense. On the other hand, we are interested in out-of-distribution generalization in this paper because LLM is supposed to encounter sentences it has not seen before. This question is inherently interesting. Out-of-distribution generalization might involve a different mechanism, but that is a separate research question.
>
> **_Question on efficacy score_**
>
> That’s a good catch. There’s a typo. We will fix it in the final version.

---

> > ### Comment · Reviewer_vgcR · 2024-08-11
> >
> > Thank you for your responses.
> >
> > Re. Limitations: I recommend including these comments in the final version of the paper.
> >
> > Re. Hijacking prompt: Thank you for conducting this quick test. I believe the paper would benefit from including a discussion on the limitations and potential side effects of hijacking using a repeated template, particularly in relation to the function of induction heads, as raised by Reviewer bm96.

---

> > > ### Author Response · Authors · 2024-08-11
> > >
> > > Thanks for your support! We will certainly incorporate these discussions into the final version of the paper. They have been very helpful in clarifying our work.

---

### Official Review · Reviewer_bm96 · 2024-07-12

**Soundness:** 3
**Presentation:** 3
**Contribution:** 2
**Rating:** 5
**Confidence:** 3

**Summary:**

This paper investigates the mechanisms underlying factual recall in transformer language models. First, the paper demonstrates a "context hijacking" phenomenon, where distractor sentences lead language models to output the wrong answer to factual questions. The paper conducts a theoretical analysis of a one-layer transformer on a noisy associative memory tasks, showing how the context hijacking phenomenon could arise. These findings are supported with experiments on synthetic data.

**Strengths:**

- The paper documents an interesting failure case for factual retrieval with LLMs ("context hijacking"). Similar "distractor" effects have been documented in prior work (which the authors cite), but I have not seen this result for the factual retrieval setting.
- The theoretical analysis presents a simple model that could give rise to the empirical phenomena, and these results are supported by a variety of experiments and analysis.
- In general, I think it is a useful contribution to provide more theoretical tools for understanding the learning dynamics of attention models, and to try to connect these analyses to real world failure cases (like context hijacking).

**Weaknesses:**

- A key argument of the paper is that an LLM can be seen as an "associative memory model", but I feel that this term lacks a precise definition. For example, one definition is that "tokens in contexts guide the retrieval of memories, even if such associations formed are not inherently semantically meaningful". It seems that the first part of this sentence would apply to any question answering model, and the notion of "not inherently semantically meaningful" needs to be defined. I think it would be especially helpful to give some examples of what an alternative model would be--associative memory, as opposed to what?
- I am not fully convinced that the one-layer transformer is a meaningful model the context hijacking phenomenon. For example, in the main example ("the Eiffel tower is not in Chicago"), it's seems like the most plausible mechanisms for either resisting context hijacking, or falling for context hijacking, would involve multi-layer Transformers--for example, the model might predict "the Eiffel tower is in Chicago" due to a kind of induction head/ICL mechanism. I think it would be helpful to expand in more detail on the connection between context hijacking and the toy model (section 5.5).

**Questions:**

- For Efficacy Score (section 3), it seems it should also require that Pr[o_] < Pr[o*] prior to modifying the context. In Fig 2a, the efficacy score seems to be the opposite of what is described in the text--the intervention makes the score go down (i.e., more often that Pr[o_] > Pr[o*]).
- In Fig 2a, are these results averaged over all relation types?
- In Section 5.4: "This implies that the self-attention layer can mitigate noise and concentrate on the informative conditional distribution π". In this setting, the final token is sampled without noise. Would this also be true if the final token had noise?

**Limitations:**

I think the authors adequately addressed the limitations of their work.

---

> ### Author Rebuttal · Authors · 2024-08-07
>
> Thank you for the support and thorough review! We’re pleased that you think the paper makes a "useful contribution”.
>
> **_Clarification on associative memory_**
>
> This is a really good question! In the literature, the definition of associative memory is usually quite broad. While any prediction can be thought of as a form of association, the key issue here is how a model can generalize beyond the associations formed by its training sets. An ideal model would understand and reason about new context sentences, compose existing knowledge and produce correct outputs. However, due to the phenomenon of context hijacking, we hypothesize that LLMs might instead rely on the appearances of certain tokens related to the output to guide memory retrieval. In this case, the association is more statistical rather than based on semantic or factual meaning. "Not semantically meaningful" in the context of fact retrieval refers to incorrect associations, as demonstrated by context hijacking examples. We will clarify this in the revised version.
>
> **_Connection to induction/ICL_**
>
> This is an excellent question. We have thought quite a bit about the connection between induction heads and ICL and cite many related works in the reference (L65-69).
>
> First, falling for context hijacking is not necessarily a problem of the induction head. In the literature, the induction head, whether it's the direct copy type [1,2] or the statistical type that relies on bigram statistics [3] requires the output token to be present in the context. However, context hijacking is slightly different. As mentioned in Section 3 (L115-116), "one can even hijack by only including words semantically close to the false target (e.g., 'France' for false target 'French')." In other words, context hijacking is about latent concept association rather than copying tokens, making it distinct from existing works on induction heads.
>
> Second, resisting context hijacking is not necessarily a property of larger models either. We have already shown that larger models like LLaMA-7B and instruction-tuned models can exhibit context hijacking in Section 3. While in some examples larger models are harder to hack (Figure 1), it is not always the case (Figure 2(b), Figure B.2(b)).
>
> Finally, in our toy model, we simplified the problem to a hypothesis-testing type problem. If the input context is modified to look like it is coming from a different distribution, then it is harder for models of different sizes to predict, though larger models would theoretically have more capacity to distinguish closer distributions. This is also not specific to the induction head mechanism.
>
> We will expand on these points more in the updated version.
>
>
> [1] Elhage, Nelson, et al. "A mathematical framework for transformer circuits." Transformer Circuits Thread 1.1 (2021): 12.
>
> [2]  Bietti, Alberto, et al. "Birth of a transformer: A memory viewpoint." Advances in Neural Information Processing Systems 36 (2024).
>
> [3] Edelman, Benjamin L., et al. "The evolution of statistical induction heads: In-context learning markov chains." arXiv preprint arXiv:2402.11004 (2024).
>
>
> **_Questions on efficacy score_**
>
> Thanks for catching the typo! It should be  Pr[o_] < Pr[o*]. On the other hand, we didn’t require Pr[o_] < Pr[o*] prior to modifying the context because we are examining data at the population level (i.e., the percentage of prompts affected). By comparing the efficacy score with no prepend (no modification) to multiple prepends, it is evident that adding misleading contexts can cause LLMs to output incorrect tokens  (for example, in Figure 2, the efficacy score drops after just one prepend).
>
> **_Fig 2a_**
>
> The results in Fig 2a are averaged across all prompts in the CounterFact dataset. We also included the standard error, although it is somewhat difficult to see.
>
> **_What if the last token is noisy_**
>
> This is a great question! The noise we refer to in section 5.4 is about occurrences of random, somewhat irrelevant tokens in the context. But we didn’t allow the last token to be sampled uniformly. Intuitively, this is because the final token is directly linked to the next token prediction and often in real sentences, not totally random. On the other hand, if the final token is sampled from the noisy distribution, then what is considered noise in the context is not that well defined anymore if the final token is from the uniform mixture. This is because the attention mechanism is to select tokens most relevant for the final token and thus randomness should be defined relative to the final token as well.

---

> > ### Comment · Reviewer_bm96 · 2024-08-13
> >
> > Thank you for responding to my questions. I will keep my score as it is (5). While I appreciate the clarifications, I still have concerns that the concept of "associative memory model" is imprecise. For me to give a higher score, I think this hypothesis needs to be defined more precisely and contrasted with some other possible explanation for the observed phenomena. Similarly, I am still not convinced that a one-layer transformer is a very useful model for thinking about context hijacking, given that most mechanisms I can think of that might be relevant in this setting would require at least two transformer layers.

---

> > > ### Author Response · Authors · 2024-08-13
> > >
> > > Thanks for your reply! Although the term “associative memory” is used loosely in the literature, we focus on a particular type of associative memory (L123-L125). To analyze this rigorously, we precisely define the latent concept association task in Sec 4.1 which concretely formalizes the idea that tokens with shared latent concepts would co-occur more (L148-184, in particular L182-184 where the final objective is defined). This is complemented by a rigorous theoretical analysis in Sec 5 (Theorems 1, 4, see also Theorem 7, 8 in App A) and detailed experiments in Sec 6. This is all in precise language that can be falsified.
> > >
> > > Furthermore, to motivate the precise task we defined, we first conducted systematic experiments on context hijacking to show that prepending the same misleading prompts more can cause LLMs to perform worse (Figure 2). This motivated us to hypothesize that LLMs might pay attention to the frequency of certain tokens in the context as supposed to understand the factual meaning of the context, which leads to the precise task we analyze in this paper.
> > >
> > > On the other hand, in this paper, we are interested in studying a **failure mode** of LLMs – context hijacking. Since we show via experiments that the failure occurs in large models like LLaMA, one can reasonably expect it to persist in smaller models as well. Because smaller models allow us to do carefully controlled experiments, it is _more meaningful_ to study how this problem persists in smaller models as a starting point, which is what we provide evidence for with latent concept association in single-layer transformers. This is different from in-context learning and induction head which are shown to work mostly for models with at least two layers.
> > >
> > >
> > > We hope this clarifies your concerns, and appreciate the opportunity to discuss these details with you.

---

### Official Review · Reviewer_rSoN · 2024-07-12

**Soundness:** 2
**Presentation:** 2
**Contribution:** 2
**Rating:** 6
**Confidence:** 4

**Summary:**

The paper presents a way to study associative memory in Transformer blocks. Specifically, the author presents a method to construct a value matrix representing associative memory and suggests its equivalence to self-attention’s value matrix. Through experiments based on synthetic data, the author proposes that the transformer gathers information through self-attention, while the value matrix stores associative memory.

**Strengths:**

1. Given the heat of LLM, and the importance of prompt engineering, the problem brought by the authors matches the field's concern. Namely, what is an important part of the architecture of LLMs that is susceptible to noise, causing errors in generation.

2. The proposed method effectively avoids the complicated effect of multi-layer attention by experimenting with a one-layer attention structure.

3. The proposed embedding’s low-rank property provides a potential way to reduce computation complexity.

**Weaknesses:**

1. Besides the findings based on pre-trained LLMs, I would like the author to dig deeper into other methods to solve context hijacking. For example, will or to what extent can supervised finetuning correct the distracted focus back to the correct context?

2. Given the prevalence of low-bit quantization, I wonder how quantization will change the associative memory. For example, after quantization, will LLM be less distracted? Or perhaps quantization will enhance the effects of the misleading context?

3. For the constructed value matrix (formula 4.1 and 5.1), even if results in section 6.1 show that using the constructed value matrix retains the accuracy, two concerns remain. First, why should a value matrix constructed from an embedding matrix be expected to act as an associative memory (intuitively)? Secondly, how and why is the constructed value matrix different from self-attention’s value matrix (gather information), if any?

4. In the results section, it is unclear how the method performs differently across different LLMs, different datasets, and other SOTA methods. It is unclear if the results can be generalized to other settings.

**Questions:**

1. In Section 5.3, it is unclear how the embeddings are trained/updated.

2. In Section 5.5, the support evidence is not sufficient. Instead of showing across different mixture rates, I would like to see how changing context (for example, adding “The Eiffel Tower is not in Chicago” to the beginning of the prompt) can potentially impact associative memory and self-attention. Another concern regarding Fig C.13 is, how/if the impact will be different should one concatenate additional context (“The Eiffel Tower is not in Chicago”) to the beginning, middle, and end of prompt.

3. In Section 6.1, Figure C.2, the author concludes that the constructed value matrix can be used to replace the self-attention value matrix without performance sacrifices. However, the author fails to provide enough justification on the consistent drop after certain dimensions, across different m. For example, when m=5, the accuracy of using the constructed matrix drops significantly after dim = 128, which hints that the constructed matrix and self-attention’s value matrix are not equivalent.

**Limitations:**

The author didn't specify any overall limitations or potential negative impact of their method. Some discussions of insufficient/abnormal behavior of the results (see the section strength/weakness and the section questions) would be helpful.

---

> ### Author Rebuttal · Authors · 2024-08-07
>
> Thanks for your thoughtful review and questions! We are glad that you find the problem of the paper matches the field's concern.
>
> **_Solving context hijacking_**
>
> We apologize if the main objective of our paper was not communicated clearly. Our primary interest lies in understanding the inner mechanisms of _pre-trained_ LLMs. To this end, our experiments on context hijacking are designed to stress test LLMs, observe their failure points, and hypothesize how LLMs work from such observations. Through these experiments, we hypothesize that LLMs might use certain tokens in contexts as clues for memory retrieval. While solving the issue of context hijacking is important, we consider it to fall outside the scope and goal of this paper.
>
> **_Quantization_**
>
> Thanks for the suggestion! Quantization is indeed a fascinating topic to explore. However, since our current paper aims to focus on understanding how LLMs achieve fact retrieval, we believe it falls outside the scope of this work. It would, however, make an excellent direction for future research.
>
>
> **_Value matrix_**
>
> This is a good question! We study a simplified one-layer transformer network. The output from self-attention is a combination of embeddings by definition. On the other hand, the unembedding (the last linear layer) is tied with the embedding (L192-L193). Therefore, intuitively, it makes sense to construct the value matrix using embeddings, as the value matrix lies between the self-attention mechanism and the unembedding layer, both of which are related to embeddings. Similar construction is also shown in other papers as well [1, 2].
>
> On the other hand, the constructed value matrix is different from the trained value matrix as we can see those accuracies after replacement do not match exactly in Fig C.2. This difference arises because the constructed value matrix is only an approximation and a simplified model, whereas the trained value matrix results from more complex training dynamics.
>
>
> [1] Bietti, Alberto, et al. "Birth of a transformer: A memory viewpoint." Advances in Neural Information Processing Systems 36 (2024).
>
> [2] Cabannes, Vivien, Elvis Dohmatob, and Alberto Bietti. "Scaling laws for associative memories." arXiv preprint arXiv:2310.02984 (2023).
>
> **_Comparison to other methods_**
>
> We apologize if we misunderstand your question, however, we have not proposed any new methods in this paper. The experiments on context hijacking serve only as a robustness test, and we conducted them across different LLMs and various prompts. There is no new method proposed in this paper.
>
>
> **_Embeddings training_**
>
> Sorry about the confusion. All the experiments on single-layer transformers, unless otherwise stated, are trained jointly with AdamW. Training details can be found in Appendix C. We will revise the paper to clarify this point earlier in the paper.
>
>
>
>
> **_Question about section 5.5_**
>
>
> Our original experiments on context hijacking do add distracting prompts like “The Eiffel Tower is not in Chicago” at the beginning of the prompt (Figure 1, L103-121). On the other hand, experiments in section 5.5 and Fig C.13 are on synthetic datasets generated from the latent concept association tasks, rather than real sentences. These experiments are designed to simulate context hijacking. We only do experiments on simulated data as opposed to real sentences in this case because it allows us to do controlled experiments like changing the mixture rate.
>
>
> **_Question about Figure C.2 in Section 6.1_**
>
> We didn’t claim that the constructed value matrices and trained value matrices are equivalent, nor should one use the construction as a method to replace trained value matrices. Rather, the goal is to show that the constructed value matrices are similar to trained value matrices (L224-L225; “it turns out empirically that this construction of Wv is _close_ to the trained Wv, even in the noisy case”) so that one can use the constructed ones to gain insight into how trained value matrices work. Indeed, even with the accuracy drops at certain dimensions, such approximation is still nontrivial compared to the baselines which are based on randomly constructed matrices composed of embeddings (see green line in Figure C.2).

---

> > ### Comment · Reviewer_rSoN · 2024-08-13
> >
> > Thank you for your responses. I will raise my evaluation.
> >
> > Re. Value matrix
> > Thank you for the clarifications. Given that those accuracies after replacement do not match exactly in Fig C.2, it is worthwhile to address this, as supposed to L337 “Figure C.2 indicates that the accuracy does not significantly decrease when the value matrix is replaced with the constructed ones”
> >
> > Re. section 5.5
> > Regarding the first half of the original comment, the question was, besides Efficacy Scores or Accuracy, have the author explored/proved a shift in attention after context hijacking? Regarding the second half, I wonder if the author has explored context hijacking to the middle, or end of the prompt?

---

> > > ### Author Response · Authors · 2024-08-14
> > >
> > > Thank you for taking the time to review our work and for adjusting your score. We greatly appreciate your effort and consideration. Regarding value matrices, we think that the constructed value matrices are only approximations. Achieving a more precise approximation could likely enhance accuracy further and is an interesting future direction.
> > >
> > > For Section 5.5, thanks for the suggestion! We haven't yet examined the attention differences before and after context hijacking, and we agree that this is a very interesting direction to explore in future work.
> > >
> > > For the second half, we have only tried putting misleading prompts at the beginning. This is because we want to keep the original query prompt intact. Inserting misleading prompts in the middle could cause the original prompt to lose its meaning. Placing them at the end would alter the original next token prediction task.

---

### Decision · Program_Chairs · 2024-09-25

**Decision:**

Accept (poster)

**Comment:**

Summary:
The paper studies the brittleness of factual recall among a variety of LLMs, where the authors show that by simply manipulating the words within the context of a prompt (without changing their semantics), one can force the model to generate wrong answers (a.k.a., context hijacking). The authors then propose a theoretical formulation of a hypothesis that could explain such a phenomenon and show its empirical applicability on single-layer transformers (the building blocks of LLMs). Specifically, they show that single-layer transformers gather information using self-attention and use the value matrix as an associative memory.

Strengths:
Understanding the workings of LLMs is an important problem, and the ML community at large would greatly benefit from it. The paper takes an important step in that direction by showing a specific failure case and unraveling its potential cause. The authors propose a simple theory and convincingly prove its validity empirically, albeit on single-layer transformer models. The paper is well-organized, well-written, and easy to understand. The theory presented is clear, and the empirical evidence is sufficient.

Weaknesses:
One weakness of the paper, which should be addressed in the final version (if accepted), is that the authors do not discuss the limitations of their work. Specifically, under what circumstances will their proposed theory fail? On a related note, for instance, the experiments are done only on single-layer transformer models. While they are indeed the building blocks of LLMs, there’s likely a lot more going on in full LLM models. The paper would be strengthened if the authors could comment on how their ideas could be generalized to explain more complex models.

Overall Recommendation:
Given the importance of the problem the paper is addressing, the soundness of the solution proposed, and the overall quality of the presentation, the reviewers unanimously recommend that this paper be accepted. The wider ML community will benefit from the dissemination of this research.